# NeRF Revisited: Fixing Quadrature Instability in Volume Rendering

**Mikaela Angelina Uy**[1]     **George Kiyohiro Nakayama**[1]     **Guandao Yang**[1,2]
**Rahul Krishna Thomas**[1]     **Leonidas Guibas**[1]     **Ke Li**[3,4]
[1]Stanford University     [2]Cornell University     [3]Simon Fraser University     [4]Google
{mikacuy, w4756677, guandao, rt03mas, guibas}@stanford.edu, keli@sfu.ca

## Abstract

Neural radiance fields (NeRF) rely on volume rendering to synthesize novel views. Volume rendering requires evaluating an integral along each ray, which is numerically approximated with a finite sum that corresponds to the exact integral along the ray under piecewise constant volume density. As a consequence, the rendered result is unstable w.r.t. the choice of samples along the ray, a phenomenon that we dub *quadrature instability*. We propose a mathematically principled solution by reformulating the sample-based rendering equation so that it corresponds to the exact integral under piecewise linear volume density. This simultaneously resolves multiple issues: conflicts between samples along different rays, imprecise hierarchical sampling, and non-differentiability of quantiles of ray termination distances w.r.t. model parameters. We demonstrate several benefits over the classical sample-based rendering equation, such as sharper textures, better geometric reconstruction, and stronger depth supervision. Our proposed formulation can be also be used as a drop-in replacement to the volume rendering equation for existing methods like NeRFs. Our project page can be found at pl-nerf.github.io.

## 1 Introduction

The advent of neural radiance fields (NeRF) [18] has sparked a flurry of work on neural rendering and has opened the way to many exciting applications [5, 11, 8, 20]. One of the key underpinnings of NeRF is volume rendering [14] – it is especially well-suited to end-to-end differentiable rendering [9], since the rendered image is a smooth function of the model parameters. This has made it possible to learn the 3D geometry and appearance solely from a 2D photometric loss on rendered images.

In volume rendering, the rendered colour $\hat{y}$ for every pixel is an expectation of the colours along the ray cast through the pixel w.r.t. the distribution over ray termination distance $s$ [6].

$$\hat{y} = \mathbb{E}_{s \sim p(s)}[c(s)] = \int_0^\infty p(s)c(s)\,\mathrm{d}s \tag{1}$$

where $p(s)$ denotes the probability density function (PDF) of the distribution over ray termination distance $s$ and $c(s)$ denotes the the colour as a function of different points along the ray.

In general, $p(s)$ and $c(s)$ can be of arbitrary forms, so evaluating this integral analytically is not possible. Therefore, in practice, $\mathbb{E}_{s \sim p(s)}[c(s)]$ is approximated with quadrature. The quadrature formula that is most commonly used in the NeRF literature takes the following form:

$$\mathbb{E}_{s \sim p(s)}[c(s)] = \sum_{j=0}^N T_j \left(1 - e^{-\tau_j(s_{j+1}-s_j)}\right) c_j, \tag{2}$$

where $T_j = \exp\left(-\sum_{k=0}^j -\tau_k(s_{k+1}-s_k)\right)$ and $\tau_k$ is the opacity evaluated at a sample $s_k$ along the ray.

This expression is derived from the exact integral under a piecewise constant assumption to the opacity and colour along the given ray [14].

37th Conference on Neural Information Processing Systems (NeurIPS 2023).

However, this seemingly simple, innocuous assumption can result in the rendered image being sensitive to the choice of samples along the ray at which the opacity $\sigma(s)$ and colour $c(s)$ are evaluated. While this does not necessarily cause a practical issue in classical rendering pipelines [14, 15, 10], it has surprising consequences when used in neural rendering. Specifically, because the opacity at all points within the interval between two samples is assumed to be the same, there is a band near the surface of the geometry where opacity at all points within the band is as high as points on the surface itself. Because different rays cast from different cameras can pass through this band at different angles and offsets, both the number and the positions of samples within this band can be very different across different rays (see Fig 1 and Fig 2 for two example scenarios). Hence, simultaneously supervising these rays to produce the same colour values as in the real image captures can give rise to conflicting supervisory signals, which can result in artifacts like fuzzy surfaces and blurry texture.

Moreover, because the piecewise constant opacity assumption gives rise to a closed form expression that is equivalent to an expectation w.r.t. a discrete random variable, it is common practice in the NeRF literature to draw samples along the ray from the discrete distribution [17]. This is commonly used to draw importance samples, and also to supervise the samples along the ray [28], for example in losses that penalize deviation of the samples from the true depth [6]. Sampling from the discrete distribution requires the definition of a continuous surrogate function to the cumulative distribution function (CDF) of the discrete random variable, which unfortunately yields imprecise samples. As a result, samples that are drawn may not be close to the surface even if the underlying probability density induced by the NeRF is concentrated at the surface. Additionally, individual supervision cannot be provided to each sample drawn from the surrogate, because the gradient of the loss w.r.t. each sample would be almost zero everywhere.

All these issues, i.e. conflicting supervision, imprecise samples and lack of supervision on the CDF from the samples, stem from the assumption that opacity is piecewise constant causing the *sensitivity to the choice of samples* both during rendering and sampling. We dub this problem as *quadrature instability*. In this paper, we revisit the quadrature used to approximate volume rendering in NeRF and devise a different quadrature formula [14] based on a different approximation to the opacity. We first show that interestingly a closed-form expression can be derived under any piecewise polynomial approximation for opacity. When the polynomial degree is 0, it reduces to the piecewise constant opacity as in existing literature, and when the degree is 2 or more, we show that it would lead to poor numerical conditioning

Therefore, we further explore a degree of 1 (i.e., piecewise *linear*) and show that it both resolves quadrature instability and has good numerical conditioning. We derive the rendering equation under *piecewise linear opacity* explicitly and show that it has a simple and intuitive form. This results in a new quadrature method for volume rendering, which can serve as a drop-in replacement for existing methods like NeRFs. We demonstrate that this reduces artifacts, improves rendering quality and results in better geometric reconstruction. We also devise a new way to sample directly from the distribution of samples along each ray induced by NeRF without going through the surrogate, which opens the way to a more refined importance sampling approach and a more effective method to supervise samples using depth.

## 2   Related Work

**NeRFs.**   Neural Radiance Field (NeRF) is a powerful representation for novel-view synthesis [18] that represents a scene using the weights of an MLP that is rendered by volumetric rendering [14]. A key finding to the success of NeRF was the use of positional encoding [26, 29] to effectively increase the capacity of the MLPs that models the opacity and emitted color as a function of a 3D coordinate and viewing direction. Many works extend NeRF such as handling larger or unbounded scenes [40, 3, 37, 25], unconstrained photo collections [13], dynamic and deformable scenes [11, 19] and sparser input views [6, 39, 31, 28]. There are a number of papers that aim to improve the rendering quality of NeRF. Some do so by utilizing different kinds of supervision such as NeRF in the Dark [16], while others tackle this by improving the model [2, 36, 4]. MipNeRF [2] changes the model input by introducing integrated positional encoding (IPE) to reduce the aliasing effect along the xy coordinates. DiVeR [36] predicts the rendered colour within a line interval directly from a trilinearly interpolated feature in a voxel-based representation ZipNeRF [4] modifies the proposal network to a grid enabling it to be used together with IPE. In contrast, our work focuses on changing the objective function by modifying the rendering equation from piecewise constant opacity to piecewise linear, while keeping the model and supervision fixed. Additionally, ZipNeRF [4] also

brings up a model specific issue on z-aliasing, where their model struggles under this setting. Similar to z-aliasing observed by ZipNeRF, we consider the setting of having conflicting supervision when presented with training views at different distances from the scene. While they may appear similar on the surface, the phenomena we study is different in that it is general and independent of the model, on having conflicting ray supervision from camera views, e.g. different camera-to-scene distances and the grazing angle setup.

**Importance Sampling on NeRFs.** Densely sampling and evaluating NeRF along multiple points in each camera ray is inefficient. Inspired by an early work on volume rendering [10], prior works typically use a coarse-to-fine hierarchical sampling strategy where the final samples are obtained by importance sampling of a coarse proposal distribution [18, 2, 3, 8]. These importance samples are drawn using inverse transform sampling where a sample is obtained by taking the inverse of the cumulative density function (CDF) of the proposal ray distribution. However, prior NeRF works that assume piecewise constant opacity result in a non-invertible CDF, and instead introduce a surrogate invertible function derived from the CDF in order to perform importance sampling. In contrast, our work that utilizes a piecewise linear opacity assumption results in an invertible CDF and a closed-form solution to obtain samples with inverse transform sampling. Other works also attempt to alter sampling using neural networks or occupancy caching [31, 23, 24, 12]. These techniques are orthogonal to our work as they propose changes to the model as opposed to our work where importance sampling is derived from a given model.

**Volume rendering.** Volume rendering is an important technique in various computer graphics and vision applications as explored in different classical works [10, 33, 7, 14]. These works include studying ray sampling efficiency [10] and data structures, e.g. octree [22] and volume hierarchy [21] for coarse-to-fine hierarchical sampling. The crux behind volume rendering is the integration over the weighted average of the color along the ray, where the weights is a function of the volume density (opacity). Max and Chen [14, 15] derive the volume rendering equation under the assumption of piecewise constant opacity and color, which NeRF [17] and its succeeding works use to learn their neural scene representation. However, the piecewise constant assumption results in rendering outputs that are sensitive to the choice of samples as well the non-invertible CDF introducing drawbacks to NeRF training. Following up on [14], works also [34] derive the volume rendering equation under the assumption that both opacity and color are piecewise linear that yield unwieldy expressions that lead to numerical issues and/or are expensive to compute. Some earlier works on rendering unstructured polygonal meshes that attempt to use this model [35], but it is in general not commonly used in practice due to the mentioned issues and hence has yet to be adopted into learning neural scene representations. In this work, we address both sets of issues that arise from the piecewise constant opacity and color and piecewise linear opacity and color by reformulating the volume rendering equation to assume piecewise linear opacity and piecewise constant color. Our derivation results in a simple and closed-form formulation to volume rendering making it suitable for NeRFs.

## 3  Background

### 3.1  Volume Rendering Review

**Definitions.** In classical literature, the process of volume rendering [15] mapping a 3D field of optical properties to a 2D image and the visual appearance is computed through the exact *integration* of these optical properties along the viewing rays. In this optical model, each point in space is an infinitesimal particle with a certain *opacity* $\tau$ that emits varying amounts of light, represented as a scalar *color* $c$, in all viewing directions. The opacity $\tau$ is the differential probability of a viewing ray hitting a particle – that is for a viewing ray $\mathbf{r}(s) = \mathbf{o} + s\mathbf{d}$, where $\mathbf{o}$ is the view origin and $\mathbf{d}$ is the ray direction, the probability of ray $\mathbf{r}$ hitting a particle along an infinitesimal interval $ds$ is $\tau(\mathbf{r}(s))ds$. Moreover, the transmittance $T_{\mathbf{r}}(s)$ is defined as the probability that the viewing ray $\mathbf{r}$ travels a distance $s$ from the view origin without terminating, i.e. without hitting any particles.

**Continuous probability distribution along the ray r.** As illustrated by Max and Chen [15], the probability of hitting a particle $s + ds$ only depends on the probability of hitting a particle at $s$ and not any particles before it the probability of ray $\mathbf{r}$ terminating at distance $s$ is given by $\tau(\mathbf{r}(s))T_{\mathbf{r}}(s)$, where $T_{\mathbf{r}}(s) = \exp(-\int_0^s \tau(\mathbf{u})du)$. Hence the continuous probability density function (PDF) of ray $\mathbf{r}(s)$, which describes the likelihood of a ray terminating and emitting at $s$, is given by

$$p(s) = \tau(\mathbf{r}(s))T_{\mathbf{r}}(s), \tag{3}$$

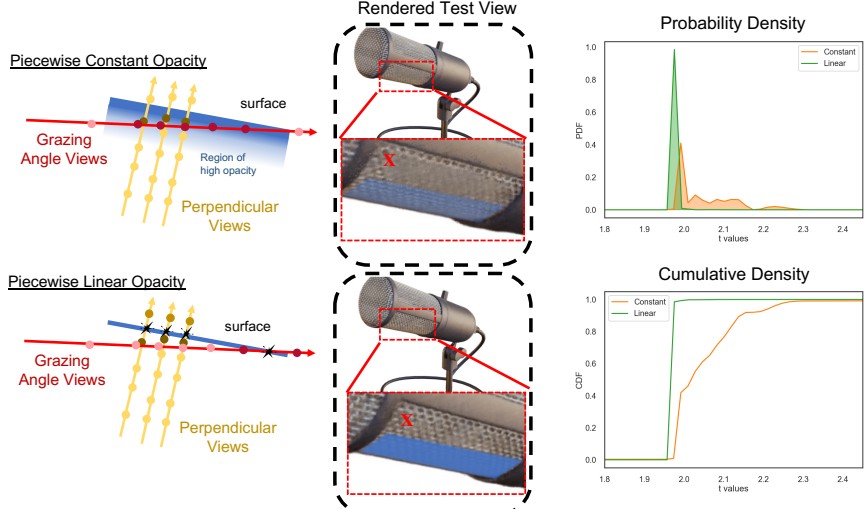

Figure 1: **Ray Conflicts: Grazing Angle.** (Left) Illustration of conflicting ray supervision at the grazing under the piecewise constant opacity. For the constant setting, to render perpendicular rays (yellow) correctly, the model has to store the associated optical properties at a region in front of the surface as a sample takes the values of the left bin boundary. In the presence of a ray near the grazing angle, it will be crossing this region of high opacity (the gradient in front of the surface), associating it with conflicting opacity/color signals. (Middle) This results in fuzzier surfaces as shown along the side of the microphone as there is a conflict in ray supervision between the perpendicular and grazing angle rays. Our piecewise linear opacity assumption alleviates this issue and results in a clearer rendered view. (Right) As shown, the resulting PDF is peakier and the CDF is sharper for our linear setting, where the plotted distributions correspond to the ray from the marked pixel in red.

where $s \in [0, \infty]$ and $\mathbf{r}(s)$ is a point on the ray $\mathbf{r}$. For notational simplicity we omit $\mathbf{r}$ and write it as $p(s) = \tau(s)T(s)$.

**Volume Rendering as a Continuous Integral.** The observed color of the ray is then the expected value of the colors $c(s)$ of all particles $s$ along the ray weighted by the probability of hitting them. Mathematically, this results in the following continuous integral[1]:

$$\mathbb{E}_{s \sim p(s)}[c(s)] = \int_0^\infty p(s)c(s)\,\mathrm{d}s = \int_0^\infty \tau(s)T(s)c(s)\,\mathrm{d}s. \tag{4}$$

**Quadrature under Piecewise Constant Opacity $\tau$.** Since this integral cannot in general be evaluated analytically, it is approximated with quadrature. Let $s_1, s_2, ..., s_N$ be $N$ (ordered) samples on the ray that define the intervals, where $I_j = [s_j, s_{j+1}]$ is the $j^{\text{th}}$ interval, and $I_0 = [0, s_1], I_N = [s_N, \infty]$. The volume density for particles along the interval $I_j$ is then approximated under the assumption that opacity is constant along each interval, making it *piecewise constant* along the ray [15]. That is, for all $j$ we have:

$$\forall s \in [s_j, s_{j+1}], \tau(s) = \tau(s_j), \tag{5}$$

for brevity we denote $\tau(s_j) = \tau_j$, i.e. $\tau_j$ is the opacity for sample $s_j$. Under this piecewise constant opacity assumption, the volume rendering equation Eq. 4 then becomes as follows:

$$\mathbb{E}_{s \sim p(s)}[c(s)] = \sum_{j=0}^N P_j c_j = \sum_{j=0}^N \left( \int_{s_j}^{s_{j+1}} \tau(u)T(u)\,\mathrm{d}u \right) c_j = \sum_{j=0}^N T_j \left( 1 - e^{-\tau_j(s_{j+1}-s_j)} \right) c_j, \tag{6}$$

---

[1]Practically the integral is taken with near $s_n$ and far $s_f$ bounds.

where $T_j = \exp\left(-\sum_{k=0}^{j} -\tau_k(s_{k+1} - s_k)\right)$. Here, color $c_j$ is also approximated to be constant along each interval $I_j$, and $P_j$ is the probability of each interval. Now, let us define the discrete random variable $\tilde{s} = \tilde{f}(s)$, where

$$\tilde{f}(x) = \begin{cases} s_0 & x \leq s_0 \\ s_j & s_j \leq x < s_{j+1} \text{ for all } j \in \{1, ..., N-1\} \\ s_N & x > s_N \end{cases}, \tag{7}$$

which gives corresponding probability mass function $\tilde{P}(s)$. Observe that the analytical expression of the integral Eq. 6 turns out to be the same as taking the expectation w.r.t. the discrete random variable $\tilde{s}$, i.e. $\mathbb{E}_{s \sim p(s)}[c(s)] = \mathbb{E}_{\tilde{s} \sim \tilde{P}(s)}[c(\tilde{s})]$. This piecewise constant opacity assumption in the volume rendering equation is used in most, if not all, existing NeRF works. We recommend the reader to read [15] for more detailed derivations and our supplementary for a more thorough walkthrough.

## 3.2 Neural Radiance Fields.

Following Max and Chen [14], Mildenhall et.al. [18] introduced neural radiance fields, a neural scene representation that uses the volume rendering equation under the piecewise constant opacity assumption for novel view synthesis. A neural radiance field (NeRF) is a coordinate-based neural scene representation, where opacity $\tau^\theta : \mathbb{R}^3 \to \mathbb{R}_{\geq 0}$ and color $c^\psi : \mathbb{R}^3 \times \mathbb{S}^2 \to [0, 255]^3$ are predicted at each continuous coordinate by parameterizing them as a neural network. To train the neural network, 2D images are used as supervision where each viewing ray is associated with a ground truth color. Volume rendering allows for the 3D coordinate outputs to be aggregated into an observed pixel color allowing for end-to-end training with 2D supervision. The supervision signal are on the coordinates of the ray samples $s_1, ..., s_N$, which updates the corresponding output opacity and color at those samples. NeRF uses a importance sampling strategy by drawing samples from the ray distribution from a coarse network to generate better samples for rendering of their fine network. To sample from a distribution, inverse transform sampling is needed, that is, one draws $u \sim U(0, 1)$ then passes it to the inverse of a cumulative distribution (CDF), i.e. a sample $x = F^{-1}(u)$, where $F$ is the CDF of the distribution. Under the piecewise constant assumption, the CDF of the discrete random variable $\tilde{s}$ is given by:

$$\tilde{F}(x) = \begin{cases} 0 & x \leq s_0 \\ \sum_{k<j} \tilde{P}(s_k) & 1 \leq x < s_{j+1} \text{ for all } j \in \{1, ..., N-1\} \\ 1 & x > s_N \end{cases}. \tag{8}$$

This CDF is however non-continuous and non-invertible. NeRF's approach to get around this is to define a surrogate invertible function $G$ derived from its CDF, then taking $x = G^{-1}(u)$. Concretely, $G(y) = \frac{y - s_{j-1}}{s_j - s_{j-1}}\tilde{F}(s_j) + \frac{s_j - y}{s_j - s_{j-1}}\tilde{F}(s_{j-1})$, where $y \in [s_{j-1}, s_j]$. However, this does not necessarily result in the samples from the actual ray distribution $p(s)$ from the model.

## 4 Drawbacks of Piecewise Constant Opacity $\tau$ in NeRFs

Unfortunately, there are properties associated with the piecewise constant opacity formulation that may not be desirable in the context of NeRFs. First, it is sensitive to the choice of samples, i.e. sample positions $s$, along the ray, a phenomenon we dub as *quadrature instability*. This quadrature instability is due to the assumption that all points in an interval take the opacity of the left bin (Eq 5), making it sensitive to sample positions. As illustrated in Figure 1, this would lead to *ray conflicts* in optimizing a NeRF when you have rays that are directly facing, i.e. perpendicular to, the surface (yellow rays) and rays that are close to the grazing angle (red ray), i.e. parallel to, the object. To render the perpendicular rays correctly, vanilla NeRF has to store the optical properties (opacity and color) associated with the perpendicular rays at a point before its intersection with the surface. This creates inaccurate signals to the optimization process when NeRF renders the ray at a grazing angle, as it will cross multiple conflicting opacity/colors (illustrated by the blue gradient). The sample sensitivity issue also arises when having cameras at different distances from the object as illustrated in Fig. 2, as this would lead to shifted sets of samples, causing inconsistencies when rendering at different camera-to-object distances. Notice that the noise on the texture of the chair is different across different viewing distances, where the middle view has fewer artifacts compared to the closer and further views.

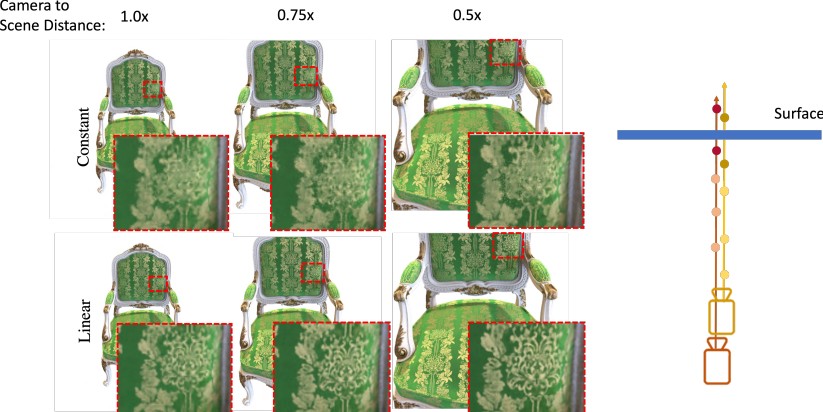

Figure 2: **Ray Conflicts: Different Camera-to-Scene Distances.** (Left) Rendered views from cameras at different distances from the object. At all distances, the rendered output for linear have sharper texture than constant because of the latter's sensitivity to the choice of samples. We also highlight the instability of the constant model as shown by the noisier texture of the middle view compared to the closer and further views. (Right) An illustration that moving the camera to different distances from the object result in different samples that lead to conflicts.

The second issue comes from the CDF $\tilde{F}$ being piecewise constant (Eq 8). This leads to two consequences. First, the piecewise constant assumption makes $\tilde{F}$ non-invertible, hence, as mentioned in the previous section, importance sampling needs to be performed via a surrogate function $G$. This results in uniformity across the samples within the bin - samples within a bin are assigned with equal probability, leading to imprecise importance samples. The second consequence comes from the fact that $\tilde{F}$ is not continuous, leading to an issue in training a NeRF that has a loss based on its samples. In other words, there will be a vanishing gradient effect when taking the gradient w.r.t. the samples, and one such example of a sample-based loss used for NeRFs is depth [28].

## 5  Generalized Form for $P_j$

We first show a generalized derivation for the probability $P_j$ of each interval $I_j$, which we use to formulate our approach that alleviates the problems described above. From $T(s) = \exp\left(-\int_0^s \tau(u)\mathrm{d}u\right)$, we first notice that:

$$\frac{\mathrm{d}T}{\mathrm{d}s} = -\exp\left(-\int_0^s \tau(u)\mathrm{d}u\right)\tau(s) = -T(s)\tau(s)$$
$$T'(s) = -T(s)\tau(s).$$

This results in the probability of each interval $I_j$ given as follows:

$$P_j = \int_{s_j}^{s_{j+1}} \tau(s)T(s)\,\mathrm{d}s = -\int_{s_j}^{s_{j+1}} T'(s)\,\mathrm{d}s = T(s_j) - T(s_{j+1}). \tag{9}$$

Since $s_j$'s are arbitrarily sampled, $P_j$ can be exactly evaluated in a closed-form expression, if and only if $T(\cdot)$ is in closed-form.

## 6  Our PL-NeRF

We observe from Eq. 15 that we can obtain a closed-form expression for $P_j$ for any piecewise polynomial function in $\tau$, which can be of any degree $d = 0, 1, 2, ..., n$. Commonly used in existing NeRF literature is choosing $d = 0$, i.e. piecewise constant, that is unstable w.r.t. the choice of samples as highlighted in the previous section. Interestingly, we also observe and show that the problem becomes numerically ill-conditioned for $d \geq 2$ making it difficult and unstable to optimize. Please see supplementary for the full proof. Hence, we propose to make opacity piecewise linear ($d = 1$), which we call **PL-NeRF**, leading to a *simple* and *closed-form* expression for the volume rendering integral that is numerically stable and is a drop-in replacement to existing NeRF-based methods. We show both theoretically and experimentally that the piecewise linear assumption is sufficient and alleviates the problems caused by quadrature instability under the piecewise constant assumption.

**Volume Rendering with Piecewise Linear Opacity.** We propose an elegant reformulation to the sample-based rendering equation that corresponds to the exact integral under **piecewise linear** opacity while keeping piecewise constant color leading to a simple and closed-form expression for the integral. That is, instead of piecewise constant opacity as in Eq 5, we assume a linear opacity for each interval $I_j$. Concretely, for $s \in [s_j, s_{j+1}]$, where $\tau_j = \tau(s_j), \tau_{j+1} = \tau(s_{j+1})$, we have

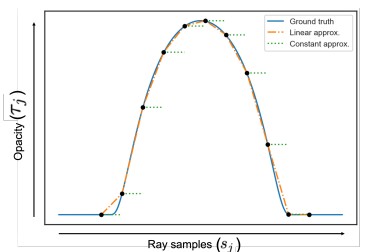

$$\tau(s) = \left( \frac{s_{j+1} - s}{s_{j+1} - s_j} \right) \tau_j + \left( \frac{s - s_j}{s_{j+1} - s_j} \right) \tau_{j+1}. \quad (10)$$

which is linear w.r.t. $s \in [s_j, s_{j+1}]$ as illustrated in Fig. 3.

Figure 3: Illustration of opacities $\tau$ along a ray under the piecewise constant (green) and piecewise linear (orange) assumptions.

Now, under the piecewise linear opacity assumption, transmittance is derived as the following closed-form expression:

$$T(s_j) = \exp\left[ -\int_0^{s_j} \tau(u)\, \mathrm{d}u \right] = \prod_{k=1}^i \exp\left[ -\int_{s_{k-1}}^{s_k} \tau(u)\, \mathrm{d}u \right],$$

$$\boxed{T(s_j) = \prod_{k=1}^i \exp\left[ -\frac{(\tau_k + \tau_{k-1})(s_k - s_{k-1})}{2} \right].} \quad (11)$$

Together with Eq. 15, this leads to the following simple and closed-form expression for $P_j$, corresponding to the exact integral under the piecewise linear opacity assumption:

$$\boxed{P_j = T(s_j) \cdot \left( 1 - \exp\left[ -\frac{(\tau_{j+1} + \tau_j)(s_{j+1} - s_j)}{2} \right] \right).} \quad (12)$$

**Precision Importance Sampling.** Moreover, it also turns out that with our piecewise linear opacity assumption, we are able to derive an exact closed-form solution for inverse transform sampling. Recall that in Sec 4, we pointed a drawback of the CDF $\tilde{F}$ being non-invertible and discontinuous under piecewise constant opacity. We show that this is alleviated in our piecewise linear setting. Concretely, given samples $s_1, ..., s_N$ resulting in interval probabilities $P_1, ..., P_N$[2] from our derivation (Eq 12), the CDF for *continuous* random variable $t$ is then given as

$$F(t) = \int_0^t p(s)\, \mathrm{d}s = \sum_{s_j < t} P_j + \int_{s_j}^t p(s)\, \mathrm{d}s = \sum_{s_j < t} P_j + \int_{s_j}^t \tau(s) T(s)\, \mathrm{d}s. \quad (13)$$

Note that unlike in piecewise constant opacity, we do not convert a continuous random variable $s$ to a discrete random variable $\tilde{s}$, thus, the resulting CDF $F$ being continuous. Now, assuming that opacity $\tau \geq 0$ everywhere, from Eq. 13 we see that $F$ is strictly increasing. Since $F$ is continuous and strictly increasing, then it is invertible.

Finally, we have our precision importance sampling, where by inverse transform sampling, we can solve for the exact sample $x = F^{-1}(u)$ for $u \sim U(0, 1)$ from the given ray distribution $p(s)$ since the CDF $F$ is invertible under piecewise linear opacity. That is, without loss of generality, let sample $u \sim U(0, 1)$ fall into the bin $u \in [C_k, C_{k+1}]$, where $C_k = \sum_{j<k} P_j$, which is equivalent to solving for $x \in [s_k, s_{k+1}]$. Reparameterizing $x = s_k + t$, where $t \in [0, s_{k+1} - s_k]$, the exact solution for sample $u$ is given by

$$\boxed{t = \frac{s_{k+1} - s_k}{\tau_{k+1} - \tau_k} \left[ -\tau_k + \sqrt{\tau_k^2 + \frac{2(\tau_{k+1} - \tau_k)\left( -\ln \frac{1-u}{T(s_k)} \right)}{(s_{k+1} - s_k)}} \right].} \quad (14)$$

Please see supplementary for full derivation. This leads to precisely sampling from the ray distribution $p(s)$ resulting in better importance sampling and stronger depth supervision.

---

[2]We note that DS-NeRF [6] shows that this will sum to 1 assuming an opaque far plane. $s_{N+1}$ would correspond to the far plane.

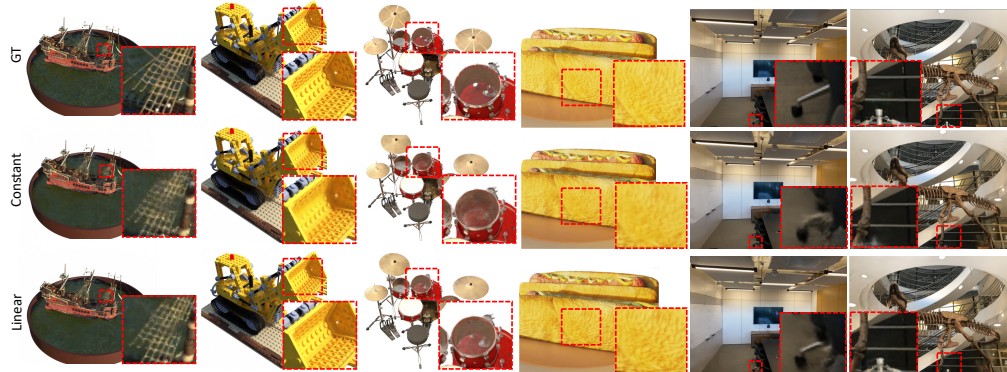

Figure 4: **Qualitative Results for Blender and Real Forward Facing**.

| | **Blender** | Avg. | Chair | Drums | Ficus | Hotdog | Lego | Mat. | Mic | Ship |
|---|---|---|---|---|---|---|---|---|---|---|
| PSNR↑ | Const. (Vanilla) | 30.61 | 32.54 | 24.79 | 29.63 | 36.08 | 32.01 | 29.31 | 32.55 | 27.95 |
| | Linear (Ours) | **31.10** | **32.92** | **25.07** | **30.18** | **36.46** | **32.90** | **29.52** | **33.08** | **28.71** |
| SSIM↑ | Const. (Vanilla) | 0.943 | 0.966 | 0.918 | 0.960 | 0.975 | 0.959 | 0.943 | 0.978 | 0.846 |
| | Linear (Ours) | **0.948** | **0.969** | **0.923** | **0.965** | **0.977** | **0.966** | **0.948** | **0.981** | **0.857** |
| LPIPS↓ | Const. (Vanilla) | 5.17 | 3.19 | 7.97 | 4.14 | 2.48 | 2.33 | 4.32 | 2.16 | 14.8 |
| | Linear (Ours) | **4.39** | **2.85** | **7.10** | **3.03** | **2.28** | **1.81** | **3.21** | **1.73** | **13.1** |
| | **LLFF** | Avg. | Fern | Flower | Fortress | Horns | Leaves | Orchid | Room | Trex |
| PSNR↑ | Const. (Vanilla) | 27.53 | 26.79 | 28.23 | 32.53 | 28.54 | 22.35 | 21.20 | 33.03 | 27.58 |
| | Linear (Ours) | **28.05** | **26.85** | **28.71** | **32.95** | **29.38** | **22.51** | **21.25** | **33.99** | **28.79** |
| SSIM↑ | Const. (Vanilla) | 0.874 | 0.746 | 0.886 | 0.925 | 0.893 | 0.816 | 0.746 | 0.956 | 0.916 |
| | Linear (Ours) | **0.885** | **0.863** | **0.902** | **0.932** | **0.911** | **0.826** | **0.754** | **0.961** | **0.933** |
| LPIPS↓ | Const. (Vanilla) | 7.37 | 9.67 | 6.34 | 2.92 | 7.26 | 11.0 | 11.8 | 4.33 | 5.66 |
| | Linear (Ours) | **6.06** | **7.92** | **4.93** | **2.46** | **5.51** | **9.59** | **10.2** | **3.54** | **4.38** |

Table 1: **Quantitative Results on Blender and LLFF Datasets.** LPIPS scores $\times 10^2$.

# 7 Results

In this section, we present our experimental evaluations to demonstrate the advantages our piecewise linear opacity formulation for volume rendering, which we call **PL-NeRF**.

## 7.1 Datasets, Evaluation Metrics and Implementation Details.

**Datasets and Evaluation Metrics.** We evaluate our method on the standard datasets: Blender and Real Forward Facing (LLFF) datasets as used in [18]. We use the released training and test splits for each. See supplementary for more details. For quantitative comparison, we follow the standard evaluation metrics and report PSNR, SSIM [32] and LPIPS [41] on unseen test views. We also report the root-mean-squared-error (RSME) on the expected ray termination in our depth experiments.

**Implementation Details.** **PL-NeRF** is implemented on top of NeRF-Pytorch [38], a reproducible Pytorch implementation of the constant (vanilla) NeRF, where we simply change the volume rendering to our formulation under piecewise linear opacity and utilize our exact importance sampling derivation. Similar to [18] we optimize a separate network for the coarse and fine models that are jointly trained with the MSE loss on ground truth images. We use a batch size of 1024 rays and a learning rate of $5 \times 10^{-4}$ that decays exponentially to $5 \times 10^{-5}$ throughout the course of optimization. We train each scene for 500k iterations which takes $\sim 21$ hours on a single Nvidia V100 GPU [3]. Our precision importance sampling enables us to use fewer samples for the fine network, hence keeping the total number of rendering samples the same, we use 128 coarse samples and 64 fine samples to train and test our method.

## 7.2 Experiments on Blender and LLFF Datasets

We first evaluate our **PL-NeRF** on the standard Blender and Real Forward Facing datasets. Table 1 shows that our **PL-NeRF** (linear) outperforms the vanilla [18] (constant) model that assumes piecewise constant opacity in all metrics for both the synthetic Blender and Real Forward Facing datasets. Figure 1 and Figure 4 show qualitative results. As shown our **PL-NeRF** is able to achieve sharper

---

[3]We rerun and train the vanilla (constant) model using the released reproducible configs from [38].

| Blender | Avg. | Chair | Drums | Ficus | Hotdog | Lego | Mat. | Mic | Ship |
|---|---|---|---|---|---|---|---|---|---|
| **PSNR↑** Mip-NeRF | 31.76 | 33.95 | 24.39 | 31.20 | 36.12 | 33.84 | 30.55 | 34.63 | 29.41 |
| PL-MipNeRF | **32.48** | **35.11** | **24.92** | **32.25** | **36.51** | **35.15** | **30.69** | **35.22** | **30.00** |
| **SSIM↑** Mip-NeRF | 0.955 | 0.975 | 0.921 | 0.971 | 0.978 | 0.971 | 0.957 | 0.987 | 0.876 |
| PL-MipNeRF | **0.959** | **0.981** | **0.928** | **0.977** | **0.980** | **0.976** | **0.959** | **0.989** | **0.882** |
| **LPIPS↓** Mip-NeRF | 3.64 | 1.80 | 6.82 | 2.35 | 1.97 | 1.44 | 2.39 | 0.973 | 11.4 |
| PL-MipNeRF | **3.09** | **1.32** | **5.78** | **1.66** | **1.67** | **1.07** | **2.09** | **0.788** | **10.3** |

Table 2: **Quantitative Results of Mip-NeRF v.s. PL-MipNeRF** LPIPS scores $\times 10^2$.

textures as shown in the Lego's scooper and the bread's surface in the hotdog. Moreover, our approach is also able to recover less fuzzy surfaces as shown in the microphone scene (Figure 1) where training views are close to the grazing angle of its head. As illustrated, the resulting probability density of the ray corresponding to the marked pixel is peakier than constant as our precision importance sampling allows us to have better samples closer to the surface. We also see clearer ropes in ship, less cloudy interior of the drum, and more solid surfaces such as cleaner leg of the swivel chair in the room scene.

### 7.3 Geometric Extraction

We also show improvement in geometric reconstruction of **PL-NeRF**. We extract the geometry from the learned density field from the trained models of PL-NeRF and Vanilla NeRF using marching cubes with a threshold of 25 following [30]. Figure 5 shows qualitative results on the reconstruction of our piecewise linear vs the original piecewise constant formulation. As shown, we are able to better recover the holes on the body and wheels of the Lego

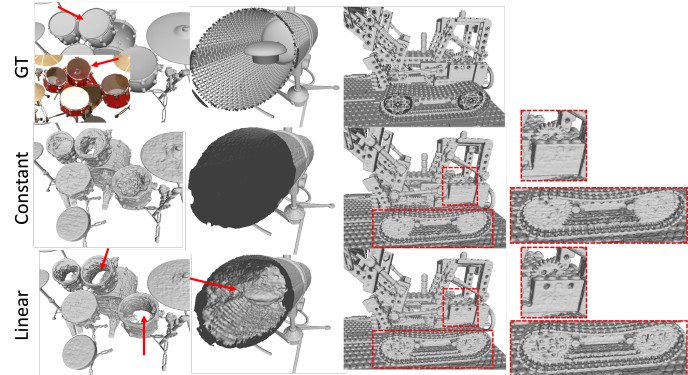

Figure 5: Geometry Extraction Qualitative Examples.

scene as well as the interior structure inside the Mic. Moreover, interestingly, the surface of the drum is reconstructed to be transparent as visually depicted in the images, as opposed to the ground truth being opaque.

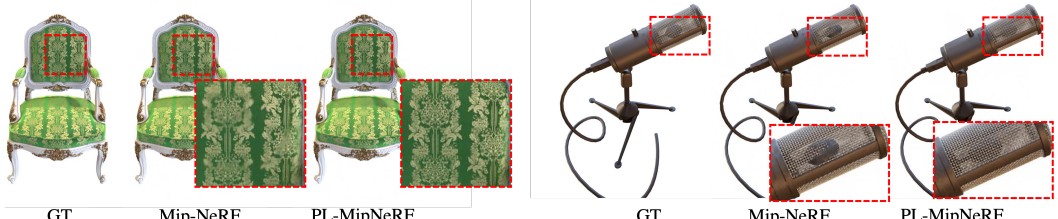

Figure 6: **Qualitative Results for Mip-NeRF vs PL-MipNeRF.** We see that under difficult scenarios such as in fine texture details of the chair and grazing angle views on the mic, PL-MipNeRF visually shows significant improvement.

### 7.4 Effectiveness of our formulation on other Radiance Field Methods

We also demonstrate our formulation's effectiveness on other radiance field methods and show that our approach can be used as a drop-in replacement to existing NeRF-based methods. We integrate our piecewise linear opacity formulation to the volume rendering integral into Mip-NeRF (**PL-MipNeRF**). Table 2 shows our quantitative results demonstrating consistent improvement across all scenes in the original hemisphere Blender dataset. Figure 6 shows qualitative examples where we see that under difficult scenarios such as when ray conflicts arise in the fine details of the Chair and in the presence of grazing angle views in the Mic, our PL-MipNeRF shows significant improvement over the baseline. Our results show that our piecewise linear opacity and piecewise constant color formulation scales well to Mip-NeRF as well. See supplementary for implementation details. We also plug our

|  | | Dist 0.25x | | | Dist 0.5x | | | Dist 0.75x | | |
|---|---|---|---|---|---|---|---|---|---|---|
| Train Set | | PSNR↑ | SSIM↑ | LPIPS↓ | PSNR↑ | SSIM↑ | LPIPS↓ | PSNR↑ | SSIM↑ | LPIPS↓ |
| **Hemisphere** | Const. (Vanilla) | 20.18 | 0.612 | 54.1 | 22.80 | 0.753 | 30.4 | 25.97 | 0.867 | 14.1 |
| | Linear (Ours) | **20.34** | **0.637** | **50.0** | **23.00** | **0.767** | **27.5** | **26.28** | **0.876** | **12.6** |
| **Multi Dist.** | Const. (Vanilla) | 22.30 | 0.677 | 45.7 | 25.51 | 0.811 | 23.1 | 28.02 | 0.891 | 11.3 |
| | Linear (Ours) | **22.66** | **0.705** | **41.1** | **26.04** | **0.828** | **20.3** | **28.55** | **0.900** | **9.90** |

Table 3: **Testing on close-up views:** *Hemisphere* are with training cameras located on the original hemisphere. *Multi Dist.* with training cameras are at random distances across a depth scale range of $0.5 - 1.0$ of the original hemisphere. Reported LPIPS score is multiplied by $10^2$.

piecewise linear opacity formulation into DIVeR [36], a voxel-based NeRF model, and show that our formulation is also an effective drop-in replacement outperforming the original DIVeR [36]. Please see supplementary for experiment results and implementation details.

## 7.5  Experiments on Close-up Views

We further consider the challenging setting of testing on cameras closer to the objects. Table 3 (top) shows quantitative results when training on the original hemisphere dataset and tested on different close-up views. As shown by the drop in metrics, the difficulty increases as the camera moves closer to the object (0.75x to 0.5x to 0.25x of the original radius) where details get more apparent. Our **PL-NeRF** outperforms the vanilla piecewise constant model in all settings and the gap (SSIM and LPIPS) between ours and the constant assumption increases as the setting becomes harder highlighting the importance of recovering shaper texture and less fuzzy surfaces.

We also consider the set-up of training with cameras with different distances to the object that result in different sets of ray samples causing conflicts. We generate training views following the data processing pipeline from [18] with a random distance scaling factor sampled from $U(0.5, 1.0)$. As shown in Table 3 (bottom), our **PL-NeRF** outperforms the vanilla constant baseline in all metrics across different camera distances, where the gap (LPIPS and SSIM) is also larger the closer the camera is to the object, where details are more apparent. The difficulty for the constant (vanilla) case under multiple camera distances is its sensitivity to the choice of samples along the ray. Figure 2 shows that the conflicting rays cause quadrature instability for under piecewise constant opacity leads to unstable outputs as shown by the noisy texture on the chair. For the constant, the level of noise (gold specs) and blurriness vary at different camera distances, whereas our **PL-NeRF** renders crisper and more consistent outputs even as the camera is moved closer or further for the object.

## 7.6  Experiments with Depth Supervision

Finally, we also show that our **PL-NeRF** enables stronger depth supervision under our piecewise linear opacity assumption due our precision importance sampling that allows for gradients to flow to these more refined samples resulting in more accurate depth. As in previous works [28], we use a sample-based loss for to incorporate depth supervision[4]. Table 4 shows quantitative

|  | PSNR↑ | SSIM↑ | LPIPS↓ | RMSE↓ |
|---|---|---|---|---|
| **Const. (Vanilla)** | 29.20 | 0.898 | 11.2 | 0.178 |
| **Linear (Ours)** | **29.54** | **0.905** | **10.4** | **0.147** |

Table 4: **Depth Supervision.** Reported LPIPS score is multiplied by $10^2$.

results when training and testing on the less constrained Blender dataset with cameras at random distances from the object, as described in the previous section, with depth supervision. As shown, our **PL-NeRF** outperforms the vanilla constant baseline on all metrics including depth RSME demonstrating that our approach allows for stronger depth supervision.

## 8  Conclusion

We proposed a new way to approximate the volume rendering integral that avoids quadrature instability, by considering a piecewise linear approximation to opacity and a piecewise constant approximation to color. We showed that this results in a simple closed-form expression for the integral that is easy to evaluate. We turned this into a new objective for training NeRFs that is a drop-in replacement to existing methods and demonstrated improved rendering quality and geometric reconstruction, more accurate importance sampling and stronger depth supervision.

---

[4]We use the original hyperparameters from [28] in this experiment.

**Acknowledgements.** This work is supported by a Apple Scholars in AI/ML PhD Fellowship, a Snap Research Fellowship, a Vannevar Bush Faculty Fellowship, ARL grant W911NF-21-2-0104, a gift from the Adobe corporation, the Natural Sciences and Engineering Research Council of Canada (NSERC), the BC DRI Group and the Digital Research Alliance of Canada.

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

# Appendix

We conduct further experiments, analysis and discussions on our proposed reformulation, where we take a piecewise *linear* approximation to opacity and piecewise *constant* approximation to color that results in an integral that is a simple and closed-form expression. This allows us to address the drawbacks of current piecewise constant assumption in NeRFs such as ray conflicts during optimization and a non-invertible CDF that lead to imprecise importance samples and vanishing gradients. We provide additional results in Sec A.1: ablation study (Sec A.1.1), a video demo (Sec A.1.2), additional results on a real dataset (Sec A.1.3), additional qualitative results (Sec A.1.4), comparison with PL-DIVeR A.1.5, additional geometric extraction results A.1.6 and comparison with less number of samples A.1.7. We then provide a walkthrough of the piecewise constant derivation from [15] (Sec A.2), which is followed by the thorough step-by-step derivation of our piecewsie linear opacity in volume rendering and precise importance sampling (Sec A.3). We also include analyses on piecewise quadratic and higher order polynomials in Sec A.4. Finally, we end with additional implementation and experiment details (Sec A.5), Limitations (Sec A.6) and Societal Impact (Sec A.7).

## A.1 Additional Results

### A.1.1 Ablation Study

We conduct a further ablation study on our precise importance sampling. As described in Sec. 4 in the main paper, our piecewise linear opacity approximation allows to solve for a closed-form solution for inverse transform sampling leading to the formulation of our **precise** importance sampling. Unlike the vanilla piecewise constant opacity approximation, our approach results in an invertible CDF, and hence we do not need to define an invertible surrogate function $G$ for inverse transform sampling as in piecewise constant opacity (see Eq. 8 of main paper), which does not necessarily result in samples from the actual ray distribution $p(s)$. We quantitatively ablate the effectiveness of our precise importance sampling (**Precise**) by replacing our formulation (Eq. 13 main paper) with the surrogate function $G$ as in the vanilla constant setting (**Surrogate**). As shown in Table A1 (first and second row of each metric), our precise importance sampling consistently outperforms using the surrogate on all metrics across all 8 scenes in the Blender dataset (hemisphere). Moreover, we also show that our precise importance sampling enables us to use fewer samples for the fine network as it is able to sample correctly from the ray distribution. Hence, keeping the same total number of rendering samples, we are able to achieve a further boost in performance by using 128 coarse and 64 fine samples as shown in Table A1 (second and third row of each metric). We use $N_c$ and $N_i$ to denote the number of coarse and fine samples, respectively.

### A.1.2 Video Demo

We also show a video demo of our results. Please see our project page (pl-nerf.github.io) for the video (best viewed in full screen). In the mic scene, we observe that the structure inside the mic is lost in the piecewise constant opacity approximation. Moreover, we see the blurriness along the sides of the body of the mic caused by ray conflicts from the perpendicular and grazing angle views during optimization of the vanilla NeRF model. For the chair scene, we see that our **PL-NeRF** is able to achieve shaper textures on the back of the chair. Moreover, we see that there is an instability to the samples as illustrated by the inconsistencies in the noise – the gold specs on the green texture (please view in full screen). This is also evident when we vary the camera-to-scene distance. The chair model was trained on multi-distance Blender data to highlight the difference when camera distances vary.

### A.1.3 Real Dataset Results

We further evaluate our **PL-NeRF** on a real dataset - DTU [1]. We train and evaluate our approach on the 15 test scenes used in [39], and report the standard metrics (PSNR, SSIM, LPIPS). We follow the protocol used in [18] for the Real Forward Facing scene where $\frac{1}{8}$ of the views were held out for testing while the rest are used for training. Table A2 shows the quantitative results averaged over the 15 scenes, where our **PL-NeRF** outperforms the constant [18] baseline.

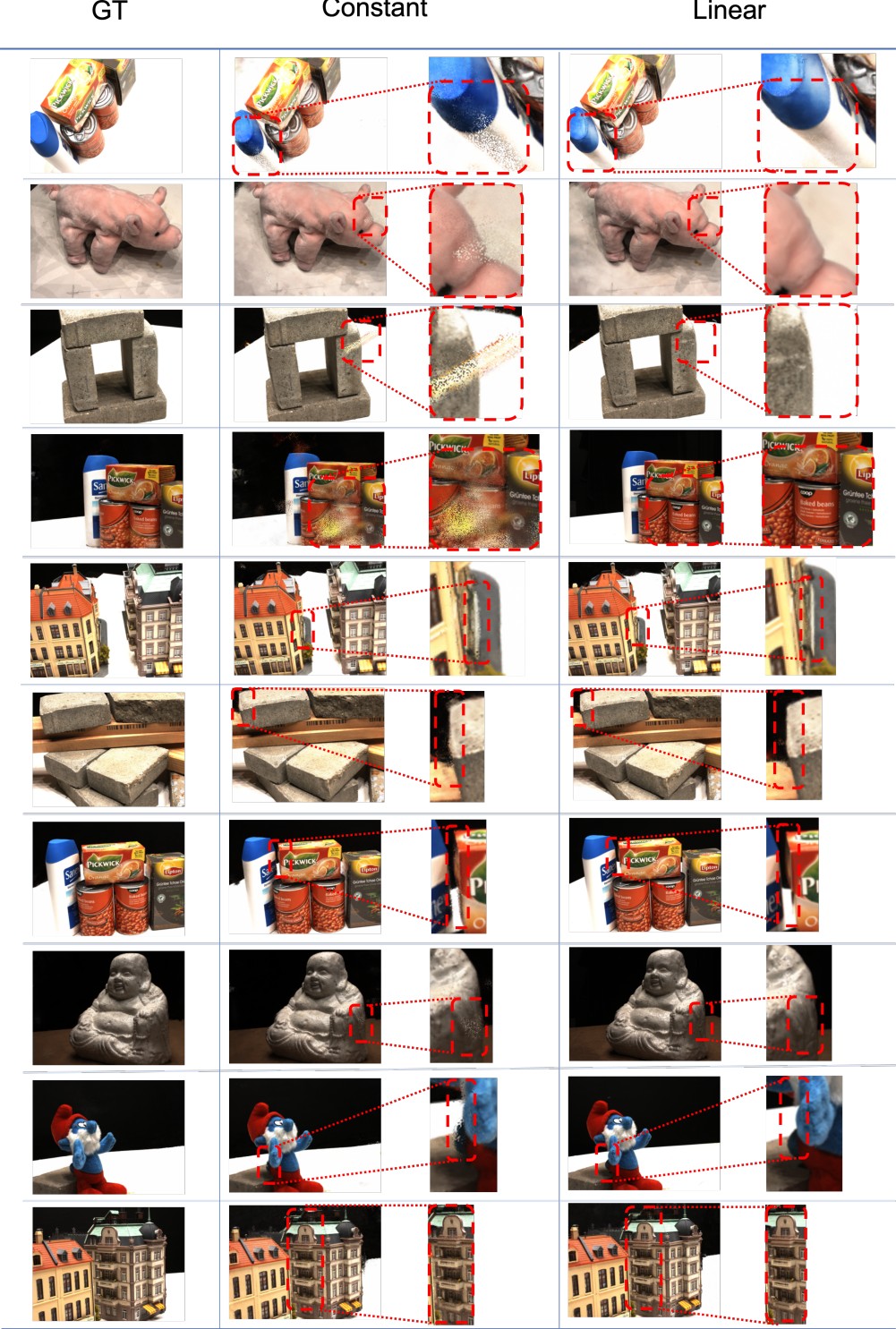

Figure A1: **DTU Qualitative Result** Visualizations of rendered DTU dataset test views. Because of the issue of grazing angle and binning inaccuracy of the piecewise constant opacity assumption, the vanilla NeRF (**constant**) exhibits blurry geometry, and rendering artifacts in the zoomed-in views (middle column). On the other hand, the piecewise linear opacity assumption in our model (**linear**) alleviates these issues (right column). Overall, the rendered views exhibit sharper surface boundaries and more faithful reconstruction compared to the constant model.

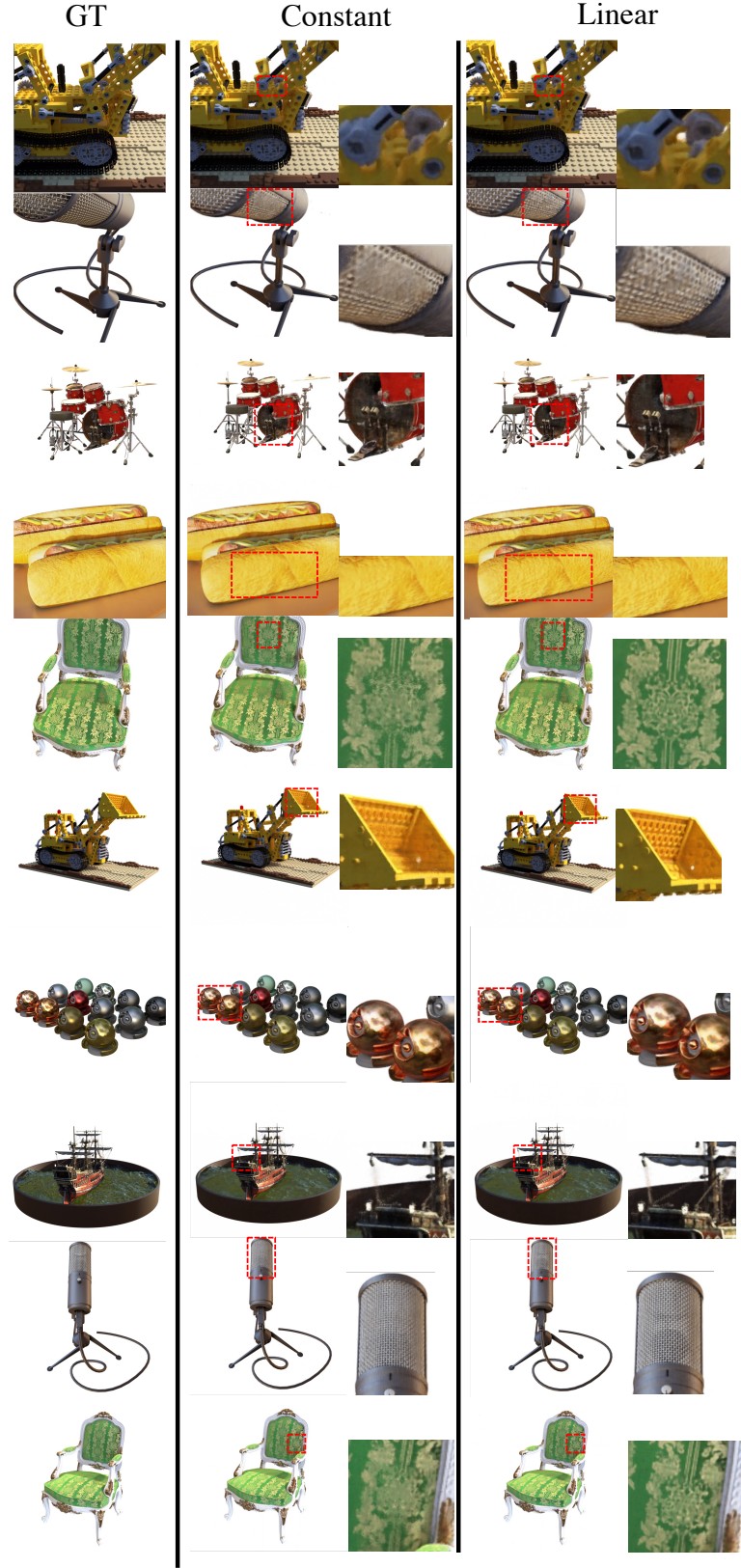

GT      Constant      Linear

Figure A2: **Blender Qualitative Result** Additonal visualizations of rendered Blender dataset test views. We see that our **PL-NeRF** is able to achieve sharper and crisper texture (chair and hotdog surface), better capture fine geometric detail (hole in lego, rope in ship) and avoid blurriness caused by conflicting rays, e.g. grazing angle views as shown in the mic.

| Metrics | Method | $N_c$ | $N_i$ | Avg. | Chair | Drums | Ficus | Hotdog | Lego | Mat. | Mic | Ship |
|---|---|---|---|---|---|---|---|---|---|---|---|---|
| PSNR↑ | Surrogate | 64 | 128 | 30.25 | 31.96 | 24.69 | 29.05 | 35.64 | 31.32 | 29.10 | 32.60 | 27.59 |
| | Precise | 64 | 128 | 30.87 | 32.85 | 24.96 | 29.61 | **36.54** | 32.27 | 29.35 | **33.21** | 28.22 |
| | Precise | 128 | 64 | **31.10** | **32.92** | **25.07** | **30.18** | 36.46 | **32.90** | **29.52** | 33.08 | **28.71** |
| SSIM↑ | Surrogate | 64 | 128 | 0.940 | 0.962 | 0.914 | 0.957 | 0.973 | 0.953 | 0.943 | 0.978 | 0.841 |
| | Precise | 64 | 128 | 0.946 | **0.969** | 0.921 | 0.961 | **0.977** | 0.962 | 0.947 | **0.982** | 0.848 |
| | Precise | 128 | 64 | **0.948** | 0.969 | **0.923** | **0.965** | 0.977 | **0.966** | **0.948** | 0.981 | **0.857** |
| LPIPS↓ | Surrogate | 64 | 128 | 5.50 | 3.85 | 8.52 | 4.15 | 2.70 | 2.94 | 4.04 | 2.30 | 15.5 |
| | Precise | 64 | 128 | 4.77 | 2.94 | 7.54 | 3.85 | **2.27** | 2.25 | 3.39 | **1.59** | 14.3 |
| | Precise | 128 | 64 | **4.39** | **2.85** | **7.10** | **3.03** | 2.28 | **1.81** | **3.21** | 1.73 | **13.1** |

Table A1: **Ablation Study.** Reported LPIPS scores are multiplied by $10^2$. We use $N_c$ and $N_i$ to denote the number of coarse and fine samples, respectively.

| | PSNR↑ | SSIM↑ | LPIPS↓ |
|---|---|---|---|
| **Const. (Vanilla)** | 27.96 | 0.909 | 8.58 |
| **Linear (Ours)** | **28.43** | **0.918** | **7.73** |

Table A2: **DTU Quantative Results** Metrics computed from the average of 15 scenes from DTU dataset. The reported LPIPS score is multiplied by $10^2$.

## A.1.4 Additional Qualitative Results

We additionally show qualitative results from DTU dataset in Fig. A1. More qualitative results are also shown in Fig. A2.

| **Blender** | | Avg. | Chair | Drums | Ficus | Hotdog | Lego | Mat. | Mic | Ship |
|---|---|---|---|---|---|---|---|---|---|---|
| PSNR↑ | DIVeR | 30.78 | 32.01 | **24.72** | 30.1 | **35.94** | 29.03 | 29.31 | 32.10 | **29.08** |
| | PL-DIVeR | **30.88** | **32.92** | 24.7 | **30.23** | 35.94 | **33.42** | **32.06** | **33.08** | 28.99 |
| SSIM↑ | DIVeR | 0.956 | 0.959 | **0.917** | **0.963** | 0.974 | 0.965 | **0.977** | 0.978 | 0.870 |
| | PL-DIVeR | **0.947** | **0.969** | 0.916 | 0.963 | **0.966** | **0.966** | 0.977 | **0.981** | **0.871** |
| LPIPS↓ | DIVeR | 3.39 | 2.79 | 6.13 | 2.34 | 1.92 | **1.46** | 1.77 | 2.16 | **7.77** |
| | PL-DIVeR | **3.28** | 2.85 | **6.01** | **2.12** | **1.83** | 1.49 | **1.77** | **1.73** | 7.82 |

Table A3: **Quantitative Results of DIVeR v.s. PL-DIVeR** Reported LPIPS scores are multiplied by $10^2$

## A.1.5 PL-DiVER

We plug our method into DIVeR by using their voxed-based representation and feature integration, and dropping in our piecewise linear opacity formulation for volume rendering (PL-DIVeR). Results are shown in Table A3 demonstrating that our approach is on-par if not better across the different scenes in the Blender dataset. We highlight that this shows the improvement of using our piecewise linear opacity formulation, which is a drop-in replacement to existing methods.

## A.1.6 Geometric Extraction

We also show quantitative results in geometric extraction improvement of PL-NeRF compared to the original Vanilla NeRF. Table A4 reports the distance between the surface of the ground truth model to the predicted meshes by sampling point clouds via ray casting. We see that our piecewise linear approach achieves a lower error compared to Vanilla NeRF on almost all the scenes in the Blender dataset.

## A.1.7 Comparison with Less Samples

We run both our PL-NeRF and Vanilla NeRF with 64 coarse and 64 fine samples results in an average of (30.09, 0.939, 0.056) and (29.86, 0.937, 0.059) for (PSNR, SSIM, LPIPS), respectively, on the Blender dataset. This shows that under less number of samples our piecewise linear opacity formulation is better than the original piecewise constant opacity assumption.

| **Blender** | | Avg. | Chair | Drums | Ficus | Hotdog | Lego | Mat. | Mic | Ship |
|---|---|---|---|---|---|---|---|---|---|---|
| CD↓ | Vanilla NeRF | 10.43 | 5.162 | **6.842** | 29.94 | 7.555 | 7.474 | 6.833 | 5.214 | 11.44 |
| | PL-NeRF | **10.10** | **4.676** | 7.754 | **29.58** | **7.004** | **6.825** | **6.061** | **5.213** | **10.44** |

Table A4: **Geometry Extraction Error** Distance between the surface of the GT to the predicted meshes. Scores are $\times 10^3$

## A.2 Volume Rendering: Walkthrough of Piecewise Constant Derivation from [15]

From [15], under the approximation that both opacity and color are piecewise constant, for $s \in [s_i, s_{i+1}]$, where $\tau_i = \tau(s_i)$ and $\tau(s) = \tau_i \forall s \in [s_i, s_{i+1}]$, the probability of the interval $P_i$ is derived as follows:

$$
\begin{aligned}
P_i &= \int_{s_i}^{s_{i+1}} \tau(s)T(s)\mathrm{d}s \\
&= \int_{s_i}^{s_{i+1}} \tau_i \exp\left(-\int_0^s \tau(u)\mathrm{d}u\right)\mathrm{d}s \\
&= \int_{s_i}^{s_{i+1}} \tau_i \exp\left(-\int_0^{s_i} \tau(u)\mathrm{d}u\right)\exp\left(-\int_{s_i}^s \tau(u)\mathrm{d}u\right)\mathrm{d}s \\
&= \int_{s_i}^{s_{i+1}} \tau_i T(s_i) \exp\left(-\int_{s_i}^s \tau_i \mathrm{d}u\right)\mathrm{d}s \\
&= \tau_i T(s_i) \int_{s_i}^{s_{i+1}} \exp\left(-\tau_i(s - s_i)\right)\mathrm{d}s \\
&= \tau_i T(s_i) \frac{\exp\left(-\tau_i(s - s_i)\right)}{-\tau_i}\Big|_{s_i}^{s_{i+1}} \\
&= \tau_i T(s_i)\Big(1 - \exp\left(-\tau_i(s_{i+1} - s_i)\right)\Big).
\end{aligned}
$$

Moreover, under the piecewise constant assumption, transmittance $T$ is derived and given by:

$$
\begin{aligned}
T(s_i) &= \exp\left(-\int_0^{s_i} \tau(u)\mathrm{d}u\right) \\
&= \prod_{j=1}^i \exp\left(-\int_{s_{j-1}}^{s_j} \tau(u)\mathrm{d}u\right) \\
&= \prod_{j=1}^i \exp\left(-\int_{s_{j-1}}^{s_j} \tau_{j-1}\mathrm{d}u\right) \\
&= \prod_{j=1}^i \exp\left(\tau_{j-1}(s_j - s_{j-1})\right).
\end{aligned}
$$

This is the formulation that is used in most, if not all, NeRF works, which has the drawbacks that we raised such as ray conflicts during NeRF optimization and non-invertible CDF causing imprecise importance sampling and vanishing gradients when defining a loss w.r.t. the samples. We propose to approach this issue by deriving the volume rendering equation under a piecewise linear approximation to opacity, which we detail in the next sections.

## A.3 Volume Rendering: Our Piecewise Linear $\tau$ Derivation

We now show our full derivation for the volume rendering equation, under the assumption that the opacity $\tau(s)$ is piecewise linear, i.e. it is linear within each interval $[s_i, s_{i+1}]$, and piecewise constant color. We then derive the probability of an interval under this assumption.

### A.3.1 Generalized form for $P_i$.

Recall the generalized form of $P_i$ as derived in the main paper. First from the definition of transmittance, we have

$$T(s) = \exp\left(-\int_0^s \tau(u)\mathrm{d}u\right)$$

$$\frac{\mathrm{d}T}{\mathrm{d}s} = -\exp\left(-\int_0^s \tau(u)\mathrm{d}u\right)\tau(s) = -T(s)\tau(s)$$

$$T'(s) = -T(s)\tau(s).$$

This results in the exact expression for the probability $P_i$ of an interval given as follows:

$$P_i = \int_{s_i}^{s_{i+1}} \tau(s)T(s)\,\mathrm{d}s = -\int_{s_i}^{s_{i+1}} T'(s)\,\mathrm{d}s = T(s_i) - T(s_{i+1}). \tag{15}$$

### A.3.2 Evaluating $\tau(s)$ for $s \in [s_i, s_{i+1}]$.

Let $\tau_j = \tau(s_i)$, $\tau_{i+1} = \tau(s_{i+1})$, be sampled points along the ray. For $s \in [s_i, s_{i+1}]$, assuming piecewise linear opacity $\tau$, i.e. $\tau(s)$ is linear within each bin, we have

$$\tau(s) = \left(\frac{s_{i+1} - s}{s_{i+1} - s_i}\right)\tau_i + \left(\frac{s - s_i}{s_{i+1} - s_i}\right)\tau_{i+1}$$

$$= \frac{1}{s_{i+1} - s_i}[(\tau_{i+1} - \tau_i)s + (s_{i+1}\tau_i - s_i\tau_{i+1})]$$

### A.3.3 Transmittance $T(s_i)$

We first derive expression for transmittance $T(s_i)$ under the piecewise linear $\tau$ assumption.

$$T(s_i) = \exp\left(-\int_0^{s_i} \tau(u)\mathrm{d}u\right)$$

$$= \prod_{j=1}^{i} \exp\left(-\int_{s_{j-1}}^{s_j} \tau(u)\mathrm{d}u\right)$$

$$= \prod_{j=1}^{i} \exp\left(\frac{-1}{s_j - s_{j-1}} \int_{s_{j-1}}^{s_j} [(\tau_j - \tau_{j-1})u + (s_j\tau_{j-1} - s_{j-1}\tau_j)]\mathrm{d}u\right)$$

$$= \prod_{j=1}^{i} \exp\left(\frac{-1}{s_j - s_{j-1}}\left[(\frac{\tau_j - \tau_{j-1}}{2})(s_j^2 - s_{j-1}^2) + (s_j\tau_{j-1} - s_{j-1}\tau_j)(s_j - s_{j-1})\right]\right)$$

$$= \prod_{j=1}^{i} \exp\left(-\left[(\frac{\tau_j - \tau_{j-1}}{2})(s_j + s_{j-1}) + (s_j\tau_{j-1} - s_{j-1}\tau_j)\right]\right)$$

$$= \prod_{j=1}^{i} \exp\left(-\frac{1}{2}\left[\tau_j s_j + \tau_j s_{j-1} - \tau_{j-1}s_j - \tau_{j-1}s_{j-1} + 2s_j\tau_{j-1} - 2s_{j-1}\tau_j\right]\right)$$

$$= \prod_{j=1}^{i} \exp\left(-\frac{1}{2}\left[\tau_j s_j - \tau_j s_{j-1} + \tau_{j-1}s_j - \tau_{j-1}s_{j-1}\right]\right)$$

Thus we get

$$\boxed{T(s_i) = \prod_{j=1}^{i} \exp\left(-\frac{(\tau_j + \tau_{j-1})(s_j - s_{j-1})}{2}\right)}. \tag{16}$$

### A.3.4 Probability of interval $[s_i, s_{i+1}]$

From the generalized form for $P_i$ as derived in the main paper, we plug in the expression for transmittance $T(s_i)$ as derived above to obtain:

$$P_i = \int_{s_i}^{s_{i+1}} \tau(s)T(s)\mathrm{d}s$$

$$= T(s_i) - T(s_{i+1})$$

$$= \prod_{j=1}^{i} \exp\left(-\frac{(\tau_j + \tau_{j-1})(s_j - s_{j-1})}{2}\right) - \prod_{j=1}^{i+1} \exp\left(-\frac{(\tau_j + \tau_{j-1})(s_j - s_{j-1})}{2}\right)$$

Hence, we obtain

$$\boxed{P_i = T(s_i) \cdot \left(1 - \exp\left[-\frac{(\tau_{i+1} + \tau_i)(s_{i+1} - s_i)}{2}\right]\right)}. \tag{17}$$

### A.3.5 Our Precision Importance Sampling

To sample from the ray distribution, inverse transform sampling is needed, that is, one draws $u \sim U(0, 1)$ then passes it to the inverse of a cumulative distribution (CDF), i.e. a sample $x = F^{-1}(u)$, where $F$ is the CDF of the distribution. Unlike the piecewise constant case, where $F$ is not invertible, needing for a surrogate function $G$ derived from F, we show that under our piecewise linear opacity assumption, we can solve for the solution $x$ for each corresponding $u$.

As illustrated in the main paper, since $F$ is continuous and increasing, under our assumption that $\tau > 0$ [5], then $F$ is invertible. Now, without loss of generality, let sample $u$ fall into the CDF interval $[c_k, c_{k+1}]$, where $c_k = \sum_{j<k} P_j$. We know that the probability of the corresponding interval is $P_k$ as given by Eq. 17. Thus we have:

$$c_{k+1} - c_k = P_k$$
$$= \int_{s_k}^{s_{k+1}} T(s)\tau(s)\mathrm{d}s$$
$$= T(s_k) \cdot \left(1 - \exp\left(-\frac{(\tau_{k+1} + \tau_k)(s_{k+1} - s_k)}{2}\right)\right)$$

We want to solve for sample $x \in [s_k, s_{k+1}]$, such that $x = F^{-1}(u)$. Equivalently, since we know that $x \in [s_k, s_{k+1}]$, then we reparameterize and let $x = s_k + t$, where $t \in [0, s_{k+1} - s_k]$. We are solving for $x$ as follows:

$$u = \int_0^x T(s)\tau(s)\mathrm{d}s$$
$$= \int_0^{s_k} T(s)\tau(s)\mathrm{d}s + \int_{s_k}^x T(s)\tau(s)\mathrm{d}s$$
$$= c_k + \int_{s_k}^x T(s)\tau(s)\mathrm{d}s$$
$$u - c_k = \int_{s_k}^x T(s)\tau(s)\mathrm{d}s$$

Now, from the derivation of the general form for $P_i$ in Eq. 15, we can similarly obtain

$$u - c_k = T(s_k) - T(x).$$
$$= T(s_k) \cdot \left(1 - \exp\left(-\int_{s_k}^x \tau(u)\mathrm{d}u\right)\right).$$

Thus, simplifying we get

$$\frac{u - c_k}{T(s_k)} = 1 - \exp\left(-\int_{s_k}^x \tau(u)\mathrm{d}u\right)$$
$$\exp\left(-\int_{s_k}^x \tau(u)\mathrm{d}u\right) = 1 - \frac{u - c_k}{T(s_k)}$$
$$-\int_{s_k}^x \tau(u)\mathrm{d}u = \ln\left(1 - \frac{u - c_k}{T(s_k)}\right)$$

which gives us the expression

$$\int_{s_k}^x \tau(u)\mathrm{d}u = \ln(T(s_k)) - \ln(T(s_k) - (u - c_k)). \tag{18}$$

---

[5]In practice, we can simply add a small $\epsilon$, say $\epsilon = 10^{-6}$, to the model output resulting in positive $\tau$, to make $\tau$ positive everywhere.

This holds for $T(s_k) \neq 0$, which is true under our assumption that $\tau > 0$, and $T(s_k) - (u - c_k) \geq 0$, which we will show in Sec. A.3.8 below.

## A.3.6 Evaluating $-\int_{s_k}^{x} \tau(u) \mathrm{d}u$.

We evaluate the expression $-\int_{s_k}^{x} \tau(u) \mathrm{d}u$ in order to solve for the exact sample $x$. Recall, for $s \in [s_k, s_{k+1}]$, we have

$$
\begin{aligned}
\tau(s) &= \left( \frac{s_{k+1} - s}{s_{k+1} - k_i} \right) \tau_k + \left( \frac{s - k_i}{s_{k+1} - s_k} \right) \tau_{k+1} \\
&= \frac{1}{s_{k+1} - s_k} [(\tau_{k+1} - \tau_k)s + (s_{k+1}\tau_k - s_k\tau_{k+1})]
\end{aligned}
$$

Let constants $a = \tau_{k+1} - \tau_k$, $b = s_{k+1}\tau_k - s_k\tau_{k+1}$, $d = \frac{1}{s_{k+1} - s_k}$, thus we can write $\tau(s)$ as follows:

$$
\tau(s) = d(as + b). \tag{19}
$$

Thus, from Eq 19 we have:

$$
\begin{aligned}
\int_{s_k}^{x} \tau(u) \mathrm{d}u &= \int_{s_k}^{x} d(au + b) \mathrm{d}u \\
&= d \left( \int_{s_k}^{x} au \, \mathrm{d}u + \int_{s_k}^{x} b \, \mathrm{d}u \right) \\
&= d \left[ \frac{au^2}{2} \Big|_{s_k}^{x} + b(x - s_i) \right] \\
&= d \left[ \frac{a(x^2 - s_k^2)}{2} + b(x - s_k) \right] \\
&= d \left[ \frac{a((s_k + t)^2 - s_k^2)}{2} + b((s_k + t) - s_k) \right] \\
&= d \left[ \frac{a(t^2 + 2s_k t)}{2} + bt \right] \\
&= d \left[ \frac{a}{2} t^2 + (as_k + b)t \right] \\
&= \frac{1}{s_{k+1} - s_k} \left[ \frac{\tau_{k+1} - \tau_k}{2} t^2 + ((\tau_{k+1} - \tau_k)s_k + s_{k+1}\tau_k - s_k\tau_{k+1})t \right] \\
&= \frac{1}{s_{k+1} - s_k} \left[ \frac{\tau_{k+1} - \tau_k}{2} t^2 + (s_{k+1}\tau_k - s_k\tau_k)t \right] \\
&= \frac{\tau_{k+1} - \tau_k}{2(s_{k+1} - s_k)} t^2 + \tau_k t
\end{aligned}
$$

Hence, plugging this in Eq. 18, we get the quadratic equation

$$
\frac{\tau_{k+1} - \tau_k}{2(s_{k+1} - s_k)} t^2 + \tau_k t - (\ln(T(s_k)) - \ln(T(s_k) - (u - c_k))) = 0.
$$

We want to solve for $t \in [0, s_{k+1} - s_k]$, and the roots of the quadratic equation is given by

$$t = \frac{(s_{k+1} - s_k)(-\tau_k \pm \sqrt{\tau_k^2 + \frac{2(\tau_{k+1}-\tau_k)(\ln T(s_k)-\ln(T(s_k)-(u-c_k)))}{(s_{k+1}-s_k)}})}{(\tau_{k+1} - \tau_k)} \tag{20}$$

That means to compute for the solution, we need to find the root

$$\frac{(-\tau_k \pm \sqrt{\tau_k^2 + \frac{2(\tau_{k+1}-\tau_k)(\ln T(s_k)-\ln(T(s_k)-(u-c_k)))}{(s_{k+1}-s_k)}})}{(\tau_{k+1} - \tau_k)} \in (0, 1)$$

which we will show always exists and is unique.

**A.3.7 Bounding** $\Delta = \tau_k^2 + \frac{2(\tau_{k+1}-\tau_k)(\ln T(s_k)-\ln(T(s_k)-(u-c_k)))}{(s_{k+1}-s_k)}$

Let us first bound the discriminant of the quadratic formula. We have

$$u - c_k \le c_{k+1} - c_k$$
$$= \sum_{j=0}^{k} P_j - \sum_{j=0}^{k-1} P_j = P_k$$
$$= T(s_k) \cdot \left(1 - \exp\left(-\frac{(\tau_{k+1}+\tau_k)(s_{k+1}-s_k)}{2}\right)\right)$$

Thus we have

$$\ln(T(s_k) - (u - c_k)) \ge \ln\left(T(s_k) - T(s_k) \cdot \left(1 - \exp\left(-\frac{(\tau_{k+1}+\tau_k)(s_{k+1}-s_k)}{2}\right)\right)\right)$$
$$= \ln\left(T(s_k) \cdot \exp\left(-\frac{(\tau_{k+1}+\tau_k)(s_{k+1}-s_k)}{2}\right)\right)$$
$$= \ln(T(s_k)) - \frac{(\tau_{k+1}+\tau_k)(s_{k+1}-s_k)}{2}$$
$$\ln T(s_k) - \ln(T(s_k) - (u - c_k)) \le \frac{(\tau_{k+1}+\tau_k)(s_{k+1}-s_k)}{2}$$

$$\frac{2(\tau_{k+1}-\tau_k)(\ln T(s_k) - \ln(T(s_k)-(u-c_k)))}{(s_{k+1}-s_k)} \le \frac{2(\tau_{k+1}-\tau_k)(\frac{(\tau_{k+1}+\tau_k)(s_{k+1}-s_k)}{2})}{(s_{k+1}-s_k)}$$
$$= \tau_{k+1}^2 - \tau_k^2$$

Hence, computing the discriminant we get:

$$\Delta = \tau_k^2 + \frac{2(\tau_{k+1}-\tau_k)(\ln T(s_k) - \ln(T(s_k)-(u-c_k)))}{(s_{k+1}-s_k)} \le \tau_k^2 + (\tau_{k+1}^2 - \tau_k^2) = \tau_{k+1}^2.$$

Similarly, $u - c_k \ge 0$, where equality holds when $u = c_k$. This gives us $\Delta \ge \tau_k^2$.

Hence, we know that $\tau_{k+1}^2 \ge \Delta \ge \tau_k^2$. Since we need

$$\frac{(-\tau_k \pm \sqrt{\Delta})}{(\tau_{k+1} - \tau_k)} \in (0, 1), \text{ and } \frac{(-\tau_k - \sqrt{\Delta})}{(\tau_{k+1} - \tau_k)} \le 0$$

Thus to find the solution $t$, we need to take the positive root. We have

$$\frac{(-\tau_k + \sqrt{\Delta})}{(\tau_{k+1} - \tau_k)} \geq \frac{(-\tau_k + \sqrt{\tau_k^2})}{(\tau_{k+1} - \tau_k)} = 0$$

$$\frac{(-\tau_k + \sqrt{\Delta})}{(\tau_{k+1} - \tau_k)} \leq \frac{(-\tau_k + \sqrt{\tau_{k+1}^2})}{(\tau_{k+1} - \tau_k)} = 1$$

This shows that the solution is within the desired interval. Hence, the solution is

$$t = \frac{(s_{k+1} - s_k)(-\tau_k + \sqrt{\tau_k^2 + \frac{2(\tau_{k+1} - \tau_k)(\ln T(s_k) - \ln(T(s_k) - (u - c_k)))}{(s_{k+1} - s_k)}})}{(\tau_{k+1} - \tau_k)} \tag{21}$$

Hence, for the positive root, we know that $t \in [0, s_{k-1} - s_k]$.

**A.3.8  Proof for $T(s_k) \geq (u - c_k)$**

$$c_k = \sum_{j=0}^{k-1} P_j$$

$$= \sum_{j=0}^{k-1} T(s_j) \cdot (1 - \exp(-\frac{(\tau_{j+1} + \tau_j)(s_{j+1} - s_j)}{2}))$$

We know that

$$T(s_j) = \prod_{i=1}^{j} \exp(-\frac{(\tau_i + \tau_{i-1})(s_i - s_{i-1})}{2})$$

Let $a_i = \exp(-\frac{(\tau_i + \tau_{i-1})(s_i - s_{i-1})}{2})$, and $T(s_0) = 1$. Hence we have

$$T(s_j) = \prod_{i=1}^{j} a_i,$$

$$c_k = T(s_0)(1 - a_1) + \sum_{j=1}^{k-1}(\prod_{i=1}^{j} a_i) \cdot (1 - a_{j+1}))$$

$$= (1 - a_1) + (a_1)(1 - a_2) + (a_1 a_2)(1 - a_3) + \ldots$$

$$= 1 - a_1 a_2 \ldots a_k$$

Since $T(s_k) = a_1 a_2 \ldots a_k$ and $a_i > 0 \forall i$ then

$$T(s_k) + c_k = a_1 a_2 \ldots a_k + (1 - a_1 a_2 \ldots a_k)$$

$$= 1$$

$$\geq c_{k+1}$$

$$\geq u. \square$$

Note that above proof and solutions hold for $\tau_k \neq \tau_{k+1}$ and $T(s_k) \neq 0$, which is all holds since we have $\tau(s) > 0 \forall s$, which is equivalent to $F$ being an increasing function.

### A.3.9 The solution for sample $u$:

Putting everything together, we have the solution $t$ given as

$$t = \frac{(s_{k+1} - s_k)(-\tau_k + \sqrt{\tau_k^2 + \frac{2(\tau_{k+1} - \tau_k)(\ln T(s_k) - \ln(T(s_k) - (u - c_k)))}{(s_{k+1} - s_k)}})}{(\tau_{k+1} - \tau_k)}. \tag{22}$$

From Sec A.3.8, we have $T(s_k) = a_1 a_2 ... a_k$ and $c_k = 1 - a_1 a_2 ... a_k$, thus $T(s_k) - (u - c_k) = 1 - u$. Hence we can simplify it to

$$t = \frac{(s_{k+1} - s_k)(-\tau_k + \sqrt{\tau_k^2 + \frac{2(\tau_{k+1} - \tau_k)(\ln T(s_k) - \ln(1 - u))}{(s_{k+1} - s_k)}})}{(\tau_{k+1} - \tau_k)}$$

$$t = \frac{(s_{k+1} - s_k)(-\tau_k + \sqrt{\tau_k^2 + \frac{2(\tau_{k+1} - \tau_k)(-\ln \frac{(1-u)}{T(s_k)})}{(s_{k+1} - s_k)}})}{(\tau_{k+1} - \tau_k)}. \tag{23}$$

## A.4 Piecewise Quadratic and Higher Order Polynomials

Now, we first consider the full derivation for volume rendering equation under the assumption that opacity is *piecewise quadratic* and color is piecewise constant. Consider opacities $\tau_1, \ldots, \tau_n$ queried at $n$ samples $s_1, \ldots, s_n$ along the ray. Here, we set $s_0 = t_n$ and $s_{n+1} = t_f$ to the near and far plane, with $\tau_0 = 0$ and $\tau_{n+1} = 10^{10}$ denoting empty and opaque space.

To interpolate opacity, because a quadratic function can only be uniquely defined with 3 points, we choose $\tau(s)$ to be quadratic within each interval $[s_j, s_{j+2}]$ for even values of $j$. To encapsulate all points, this forces $n$ to be *odd*, e.g. using $127$ coarse samples and $64$ fine samples.

### A.4.1 Derivation: Computing $T(s)$

In the same way as Sec. A.3.3, we can derive transmittance, which is in closed-form. The only modification is that the formulae for integrals of opacity are different over left and right subintervals $[s_j, s_{j+1}]$ and $[s_{j+1}, s_{j+2}]$, with $j$ even (see the next section for a derivation of these integrals):

$$T(s_{2i}) = \exp\left(-\int_{s_0}^{s_{2i}} \tau(u)du\right) = \prod_{j=1}^{i} \exp\left(-\int_{s_{2j-1}}^{s_{2j}} \tau(u)du\right) \exp\left(-\int_{s_{2j}}^{s_{2j+1}} \tau(u)du\right). \tag{24}$$

Substituting the expressions in Eqs. 28 and 29 gives a closed form expression in terms of the $s_j$'s and $\tau_j$'s. We can similarly compute

$$T(s_{2i+1}) = \exp\left(-\int_{s_{2i}}^{s_{2i+1}} \tau(u)du\right) \prod_{j=1}^{i} \exp\left(-\int_{s_{2j-1}}^{s_{2j}} \tau(u)du\right) \exp\left(-\int_{s_{2j}}^{s_{2j+1}} \tau(u)du\right). \tag{25}$$

Then the probability of the $i$th interval for each $0 \leq i \leq n$ is, as before,

$$P_i = T(s_i) - T(s_{i+1}). \tag{26}$$

This leads to a closed-form expression for $P_i$. This means the behavior of $P_i$ depends entirely on that of the opacity integral. However, due to the form of $\int \tau(s)ds$ detailed in the next section, this means $P_i$ involves a *piecewise exponential of a rational function in $\tau_j$'s and $s_j$'s*, which leads to poor numerical conditioning and thus optimization instability.

### A.4.2 Derivation: Computing and Integrating $\tau(s)$ on Intervals

We now compute $\tau(s)$ and its integral on each interval. Fix the interval $[s_j, s_{j+2}]$, with $j$ odd, and $\tau_j = \tau(s_j)$, $\tau_{j+1} = \tau(s_{j+1})$, $\tau_{j+2} = \tau(s_{j+2})$. By *Lagrange interpolation*, the quadratic $\tau(s)$ passing through $(s_j, \tau_j), (s_{j+1}, \tau_{j+1}), (s_{j+2}, \tau_{j+2})$ is given by:

$$\tau(s) = \frac{\tau_j}{\alpha_j \gamma_j}(s - s_{j+1})(s - s_{j+2}) - \frac{\tau_{j+1}}{\alpha_j \beta_j}(s - s_j)(s - s_{j+2}) + \frac{\tau_{j+2}}{\beta_j \gamma_j}(s - s_j)(s - s_{j+1}),$$
$$\alpha_j = s_{j+1} - s_j, \quad \beta_j = s_{j+2} - s_{j+1}, \quad \gamma_j = s_{j+2} - s_j. \tag{27}$$

Note the integrals of the three monic quadratics over $[s_j, s_{j+1}]$ can be expressed in terms of $\alpha_j, \beta_j, \gamma_j$:

$$\int_{s_j}^{s_{j+1}} (s - s_{j+1})(s - s_{j+2})\mathrm{d}s = \int_{-\alpha_j}^{0} s(s - \beta_j)\mathrm{d}s = \frac{\alpha_j^3}{3} + \frac{\alpha_j^2 \beta_j}{2},$$

$$\int_{s_j}^{s_{j+1}} (s - s_j)(s - s_{j+2})\mathrm{d}s = \int_{0}^{\alpha_j} s(s - \gamma_j)\mathrm{d}s = \frac{\alpha_j^3}{3} - \frac{\alpha_j^2 \gamma_j}{2},$$

$$\int_{s_j}^{s_{j+1}} (s - s_j)(s - s_{j+1})\mathrm{d}s = \int_{0}^{\alpha_j} s(s - \alpha_j)\mathrm{d}s = -\frac{\alpha_j^3}{6}.$$

Thus, using Eq. 27 gives the integral of opacity over $[s_j, s_{j+1}]$:

$$\int_{s_j}^{s_{j+1}} \tau(s)\mathrm{d}s = \frac{\tau_j}{\gamma_j} \cdot \left[\frac{\alpha_j^2}{3} + \frac{\alpha_j \beta_j}{2}\right] - \frac{\tau_{j+1}}{\beta_j}\left[\frac{\alpha_j^2}{3} - \frac{\alpha_j \gamma_j}{2}\right] + \frac{\tau_{j+2}}{\beta_j \gamma_j} \cdot \left[-\frac{\alpha_j^3}{6}\right]. \tag{28}$$

Similarly, the integrals of those same three quadratics over $[s_{j+1}, s_{j+2}]$ factor out $\beta_j$:

$$\int_{s_{j+1}}^{s_{j+2}} (s - s_{j+1})(s - s_{j+2})\mathrm{d}s = \int_{0}^{\beta_j} s(s - \beta_j)\mathrm{d}s = -\frac{\beta_j^3}{6},$$

$$\int_{s_{j+1}}^{s_{j+2}} (s - s_j)(s - s_{j+2})\mathrm{d}s = \int_{-\beta_j}^{0} (s + \gamma_j)s\mathrm{d}s = \frac{\beta_j^3}{3} - \frac{\beta_j^2 \gamma_j}{2},$$

$$\int_{s_{j+1}}^{s_{j+2}} (s - s_j)(s - s_{j+1})\mathrm{d}s = \int_{0}^{\beta_j} (s + \alpha_j)s\mathrm{d}s = \frac{\beta_j^3}{3} + \frac{\beta_j^2 \alpha_j}{2}.$$

Then, summing these up along Eq. 27 gives the integral of opacity over $[s_{j+1}, s_{j+2}]$:

$$\int_{s_{j+1}}^{s_{j+2}} \tau(s)\mathrm{d}s = \frac{\tau_j}{\alpha_j \gamma_j} \cdot \left[-\frac{\beta_j^3}{6}\right] - \frac{\tau_{j+1}}{\alpha_j}\left[\frac{\beta_j^2}{3} - \frac{\beta_j \gamma_j}{2}\right] + \frac{\tau_{j+2}}{\gamma_j} \cdot \left[\frac{\beta_j^2}{3} + \frac{\beta_j \alpha_j}{2}\right]. \tag{29}$$

Observe in Eqs. 28 and 29 that the integral of opacity involves some terms with $\alpha_j, \beta_j, \gamma_j$ in the denominator, which *do not cancel*. Thus, the integral of opacity over an interval *is not a polynomial in $\tau_i$'s and $s_i$'s*, but is instead a *rational function*. This contrasts with the linear derivation in Sec. A.3.3, where this integral was a degree 2 multivariate polynomial in $\tau_i$'s and $s_i$'s. This caveat causes numerical instability, which will be discussed further in Secs. A.4.3 and A.4.4.

Generally, following the steps of the above derivation shows that if we interpolate $\tau$ piecewise by *any degree $d$ polynomial, $d \geq 2$,* then the result is a rational function in $s_i$'s and $\tau_i$'s, but not a polynomial, which would also lead to training instability as in the quadratic case.

### A.4.3 Piecewise Quadratic Problem 1: Negative Interpolated Opacity

As seen above, one problem with the piecewise quadratic model is that the integral of opacity is a *rational function in* $\alpha_i, \beta_i, \gamma_i$. In particular, due to the presence of negatives in front of certain rational terms in Eqs. 28 and 29, it may become *negative* as the denominators of these terms approach zero. One example is shown in Figure A.4.3, for samples $s_1, s_2, s_3$ with $\tau_1 = \tau_2 < \tau_3$ and $s_3 - s_2 \ll s_2 - s_1$, where the interpolated quadratic dips far below the $x$-axis. Note this interpolation is physically implausible, as *opacity should be nonnegative everywhere*.

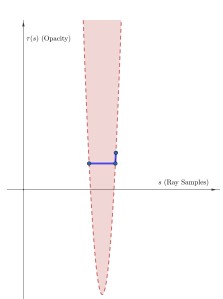

Figure A3: Interpolation gives negative $\tau$-values when $s$-values are close.

Furthermore, from Eqs. 24 and 25, when the opacity integral is negative on intervals, transmittance can then be a product with exponentials of negative terms, and potentially be greater than 1. This is incompatible with the physical interpretation of transmittance as a *probability* that a light travels a distance along a ray without being absorbed.

### A.4.4 Piecewise Quadratic Problem 2: Instability from Sample Proximity

In Eqs. 28 and 29, we observe the presence of terms such as $\alpha_j, \beta_j, \gamma_j$ in the denominator. *These do not appear in the piecewise linear model.* As a result, because these quantities approach zero as samples $s_j, s_{j+1}, s_{j+2}$ become closer, then the integral of opacity can become an arbitrarily large positive or even *negative* (as shown qualitatively in the previous section). Referencing the transmittance formulae in Eqs. 24 and 25, this means transmittance can approach zero or infinity and gradients can explode as samples are clustered together. This is fairly common in stratified sampling, and even more so with importance sampling.

To formally describe this instability, we first look at what happens when $s_j$ and $s_{j+1}$ coincide:

$$\lim_{s_{j+1} \to s_j^+} \int_{s_j}^{s_{j+1}} \tau(s) \mathrm{d}s = 0$$

because $\tau : \mathbb{R}_{\geq 0} \to \mathbb{R}$ is continuous. There is no instability in this integral when $s_{j+1} \to s_j^+$. However, when $s_{j+1}$ and $s_{j+2}$ coincide, because $\beta_j \to 0^+$ and $\gamma_j \to \alpha_j^+$, the integral approaches

$$
\begin{aligned}
\lim_{s_{j+2} \to s_{j+1}^+} \int_{s_j}^{s_{j+1}} \tau(s)\mathrm{d}s &= \lim_{s_{j+2} \to s_{j+1}^+} \left[ \frac{\tau_j}{\gamma_j} \cdot \left[ \frac{\alpha_j^2}{3} + \frac{\alpha_j \beta_j}{2} \right] - \frac{\tau_{j+1}}{\beta_j} \left[ \frac{\alpha_j^2}{3} - \frac{\alpha_j \gamma_j}{2} \right] + \frac{\tau_{j+2}}{\beta_j \gamma_j} \cdot \left[ -\frac{\alpha_j^3}{6} \right] \right] \\
&= \frac{\tau_j \alpha_j}{3} - \lim_{s_{j+2} \to s_{j+1}^+} \frac{\frac{\tau_{j+1} \alpha_j^2}{3} - \frac{\tau_{j+1} \alpha_j \gamma_j}{2} + \frac{\tau_{j+2} \alpha_j^3}{6 \gamma_j}}{\beta_j} \\
&= \frac{\tau_j \alpha_j}{3} - \lim_{s_{j+2} \to s_{j+1}^+} \frac{2\tau_{j+1} \alpha_j^2 \gamma_j - 3\tau_{j+1} \alpha_j \gamma_j^2 + \tau_{j+2} \alpha_j^3}{6 \gamma_j \beta_j} \\
&= \frac{\tau_j \alpha_j}{3} - \lim_{s_{j+2} \to s_{j+1}^+} \frac{\tau_{j+1} \alpha_j (\alpha_j^2 + 2\alpha_j \gamma_j - 3\gamma_j^2) + \alpha_j^3 (\tau_{j+2} - \tau_{j+1})}{6 \gamma_j \beta_j} \\
&= \frac{\tau_j \alpha_j}{3} - \lim_{s_{j+2} \to s_{j+1}^+} \frac{\tau_{j+1} \alpha_j (\alpha_j + 3\gamma_j)(\alpha_j - \gamma_j) + \alpha_j^3 (\tau_{j+2} - \tau_{j+1})}{6 \gamma_j \beta_j} \\
&= \frac{\tau_j \alpha_j}{3} - \lim_{s_{j+2} \to s_{j+1}^+} \frac{-\tau_{j+1} \alpha_j (\alpha_j + 3\gamma_j) \beta_j + \alpha_j^3 (\tau_{j+2} - \tau_{j+1})}{6 \gamma_j \beta_j}
\end{aligned}
$$

$$= \frac{\tau_j \alpha_j}{3} + \frac{2\tau_{j+1} \alpha_j}{3} - \lim_{s_{j+2} \to s_{j+1}^+} \frac{\alpha_j^3 (\tau_{j+2} - \tau_{j+1})}{6\gamma_j \beta_j}$$

$$= \frac{\tau_j \alpha_j}{3} + \frac{2\tau_{j+1} \alpha_j}{3} - \frac{\alpha_j^2}{6} \lim_{s_{j+2} \to s_{j+1}^+} \frac{\tau_{j+2} - \tau_{j+1}}{\beta_j}.$$

The behavior of this integral limit depends on the last limit. To analyze this, let $h : \mathbb{R}_{\geq 0} \to \mathbb{R}$ be the network function which takes in samples on the ray and outputs opacity. The last limit is

$$\lim_{s_{j+2} \to s_{j+1}^+} \frac{\tau_{j+2} - \tau_{j+1}}{\beta_j} = \lim_{s_{j+2} \to s_{j+1}^+} \frac{h(s_{j+2}) - h(s_{j+1})}{s_{j+2} - s_{j+1}}. \tag{30}$$

*Because $h$ is almost always differentiable at $s_{j+1}$, this becomes $h'(s_{j+1})$, and so*

$$\lim_{s_{j+2} \to s_{j+1}^+} \int_{s_j}^{s_{j+1}} \tau(s) \mathrm{d}s = \frac{\tau_j \alpha_j}{3} + \frac{2\tau_{j+1} \alpha_j}{3} - \frac{\alpha_j^2}{6} h'(s_{j+1}). \tag{31}$$

There are no constraints on the value of $h'(s_{j+1})$ as the network trains, so the above limit can achieve any real value. In particular, we can derive a condition for when the limit is negative:

$$\lim_{s_{j+2} \to s_{j+1}^+} \int_{s_j}^{s_{j+1}} \tau(s) \mathrm{d}s < 0 \iff h'(s_{j+1}) > \frac{2\tau_j + 4\tau_{j+1}}{\alpha_j}. \tag{32}$$

That is, whenever there is a sharp enough increase in opacity (which can happen at, say, a surface crossing) and samples $s_{j+1}, s_{j+2}$ are sufficiently close, the interpolated quadratic can have a negative integral on $[s_j, s_{j+1}]$, which leads to the issues described in Sec. A.4.3. *In other words, the integral of opacity over $[s_j, s_{j+1}]$ is unstable as $s_{j+2} \to s_{j+1}^+$.* A similar analysis holds to show the instability of the opacity integral on $[s_{j+1}, s_{j+2}]$ as $s_{j+1} \to s_j^+$.

### A.4.5   Piecewise Quadratic Problem 3: Importance Sampling

Suppose we wish to importance sample in the same manner as piecewise linear, that is, we use inverse transform sampling. In essence, we draw $u \sim U(0, 1)$ and then sample $x = F^{-1}(u)$, with $F$ the CDF of the distribution. Recall $F$ is computed as

$$f(x) = \int_{s_0}^{x} T(s)\tau(s)ds = \int_{s_0}^{s_k} T(s)\tau(s)ds + \int_{s_k}^{x} T(s)\tau(s)ds = c_k + \int_{s_k}^{x} T(s)\tau(s)ds. \tag{33}$$

As before, the latter equation becomes

$$f(x) = c_k + T(s_k) - T(x). \tag{34}$$

Hence, computing $x$ amounts to solving the equation $f(x) = u$, which becomes from above:

$$T(x) = c_k + T(s_k) - u. \tag{35}$$

As $T$ is the exponential of a cubic, this amounts to solving a cubic above. This does have a real solution, as $F$ is increasing and continuous, so the Intermediate Value Theorem implies $f(x) = u$ has a solution; and the solution is unique because $F$ is strictly increasing, as it is an exponential of a polynomial. *However, the complexity of exact importance sampling would be large.*

This analysis reveals another downside of higher order polynomials. In general, suppose we wish to interpolate opacity $\tau$ with a piecewise degree $n$ polynomial. Then following the same method as above, we see transmittance $T$ is the piecewise exponential of a degree $n + 1$ polynomial, which is the integral of $\tau$. So inverse transform sampling reduces to solving a degree $n + 1$ polynomial. The *Abel-Ruffini Theorem* asserts for $n + 1 \geq 5$ that this polynomial is in general not solvable by radicals. In other words, there is no simple closed form for exact importance sampling when $n \geq 4$.

Theoretically, this could be exactly solved for $n = 2$ and $n = 3$ (i.e. when opacity is piecewise quadratic or cubic, respectively), but the formulae to derive cubic and quartic solutions can become sufficiently complicated and the resulting complicated expressions may result in numerical instability during optimization, especially when taking the gradient w.r.t. the samples.

### A.5 Experiment details, Reproducibility and Compute

#### A.5.1 Implementation Details

We include the core code snippets of our implementation of **PL-NeRF**. Figure A4 shows the volume rendering equation that includes the implementation of Eq. 10 and 11 (main paper). This is a direct replacement of the original constant approximation, where we also show the code snippet in the figure as reference. Figure A5 shows the implementation of our precise importance sampling from Eq. 13 (main paper), which is also a direct replacement of the constant importance sampling implementation (Figure A6) for reference. We highlight that our formulation is a direct replacement of the functions from the original implementation, and hence for the depth experiments, we are also able to directly adapt the codebase from [28]. We use Nvidia v100 and A5000 GPU's for our experiments. Each scene is trained on a single GPU and takes $15 - 20$ hours. We used an internal academic cluster and cloud compute resources to train and evaluate our models.

For the MipNeRF-based experiments, our experiments are also run on the standard train and test split of the Blender dataset with the official released hyperparameters of Mip-NeRF using the NerfStudio [27] codebase. For PL-MipNeRF, we use the two-MLP training scheme with a coarse loss weight of 1.0.

For DIVeR-based experiments, we use their official implementation and configuration for DIVeR64 at 128 voxels. For PL-DIVeR, we utilize their voxel-based representation and feature integration and dropping in our piecewise linear opacity rendering formulation. We similarly run the DIVeR models on a single Nvidia v100 GPU trained using their default configurations and hyperparameters for DIVeR64 at 128 voxels.

The total training time for 500k iterations on a single Nvidia v100 GPU is 17.78 and 21.43 hours for Vanilla NeRF and PL-NeRF, respectively. Figure 2-c shows the head-to-head comparison of training PSNR (y-axis) with respect to time (x-axis) of Vanilla NeRF vs PL-NeRF on the Lego scene. Rendering a single 800x800 image takes 25.59 and 32.35 seconds for Vanilla NeRF and PL-NeRF, respectively.

#### A.5.2 Computational complexity under different number of samples

We measured the total rendering time for a single 800x800 image under different numbers of samples for our PL-NeRF. The total rendering time for (64+64), (64+128) and (128+64) are 19.20, 25.85 and 32.35 seconds, respectively.

#### A.5.3 Convergence plots under different number of samples

Figure **??** shows the convergence plots under different numbers of samples of our PL-NeRF vs Vanilla NeRF. We see that under different number of samples, our linear approach converges to a higher training PSNR.

### A.6 Limitations

Our piecewise linear opacity approximation is able to handle arbitrarily small opacity, e.g. $1e^{-6}$, however, we cannot handle coordinates with exactly zero opacity. This special case is not an issue in practice. Also theoretically, any atom absorbs light and thus will not have exactly zero opacity, except in a vacuum. Another limitation is our method is slightly slower than the original piecewise constant approximation, and this is due to requiring more FLOPS for our importance sampling computation (Eq. 13 main paper). Another additional limitation is we still assume piecewise color, i.e. within a bin we do not handle color integration. Modeling this can potentially handle difficult scenarios such as double-walled colored glass or atmospheric effects such as fog or smoke. Lastly, we also inherit the limitations of NeRFs in general, such as requiring known camera poses.

### A.7 Broader Impact

Our models require the usage of GPUs both in training time and rendering time, and GPUs use up energy to run and power them. We acknowledge that this contributes to climate change that is an important societal issue. Despite this, we observe the improvement in the results that is theoretically

```python
### Our Piecewise Linear Opacity Reformulation
def compute_weights_piecewise_linear(raw, z_vals, near, far, rays_d, noise=0., return_tau=False):
    raw2expr = lambda raw, dists: torch.exp(-raw*dists)

    dists = z_vals[...,1:] - z_vals[...,:-1]
    dists = dists * torch.norm(rays_d[...,None,:], dim=-1)

    ### tau(near) = 0, tau(far) = very big (will hit an opaque surface)
    tau = torch.cat([torch.ones((raw.shape[0], 1), device=device)*1e-10, raw[...,3] + noise, torch.ones((raw.shape[0], 1), device=device)*1e10], -1)

    tau = F.relu(tau) + 1e-6 ## Make positive

    interval_ave_tau = 0.5 * (tau[...,1:] + tau[...,:-1])

    '''
    Evaluating exp(-0.5 (tau_{i+1}+tau_i) (s_{i+1}-s_i) )
    '''
    expr = raw2expr(interval_ave_tau, dists)  # [N_rays, N_samples+1]

    ### Transmittance until s_n
    T = torch.cumprod(torch.cat([torch.ones((expr.shape[0], 1), device=device), expr], -1), -1) # [N_rays, N_samples+2], T(near)=1, starts off at 1

    ### Factor to multiply transmittance with
    factor = (1 - expr)
    weights = factor * T[:, :-1] # [N_rays, N_samples+1]

    else:
        return weights

### Original Piecewise Constant Opacity
def compute_weights(raw, z_vals, rays_d, noise=0.):
    raw2alpha = lambda raw, dists, act_fn=F.relu: 1.-torch.exp(-act_fn(raw)*dists)

    dists = z_vals[...,1:] - z_vals[...,:-1]
    dists = torch.cat([dists, torch.full_like(dists[...,:1], 1e10, device=device)], -1)  # [N_rays, N_samples]
    dists = dists * torch.norm(rays_d[...,None,:], dim=-1)

    alpha = raw2alpha(raw[...,3] + noise, dists)  # [N_rays, N_samples]
    weights = alpha * torch.cumprod(torch.cat([torch.ones((alpha.shape[0], 1), device=device), 1.-alpha + 1e-10], -1), -1)[:, :-1]
    return weights
```

Figure A4: **Code snippet for volume rendering.** The implementation for our piecewise linear opacity approximation is a drop-in replacement from the original piecewise constant.

grounded and believe that it is beneficial for the pursuit of science. We responsibly ran our models by first prototyping on selected scenes before scaling up to different scenes across datasets to minimize its impact to climate change.

```
### Our Piecewise Linear Opacity Reformulation
def sample_pdf_reformulation(bins, weights, tau, T, near, far, N_samples, det=False, pytest=False, \
    zero_threshold = 1e-4, epsilon_=1e-3):
    ### bins = z_vals, ie bin boundaries, input does not include near and far plane yet
    ### weights is the PMF of each bin ## N_samples + 1

    bins = torch.cat([near, bins, far], -1)

    pdf = weights

    cdf = torch.cumsum(pdf, -1)
    cdf = torch.cat([torch.zeros_like(cdf[...,:1]), cdf], -1)  # (batch, len(bins))

    ### CDF sums up to 1.0
    cdf[:,-1] = 1.0

    # Take uniform samples
    if det:
        u = torch.linspace(0., 1., steps=N_samples, device=bins.device)
        u = u.expand(list(cdf.shape[:-1]) + [N_samples])
    else:
        u = torch.rand(list(cdf.shape[:-1]) + [N_samples], device=bins.device)

    # Pytest, overwrite u with numpy's fixed random numbers
    if pytest:
        np.random.seed(0)
        new_shape = list(cdf.shape[:-1]) + [N_samples]
        if det:
            u = np.linspace(0., 1., N_samples)
            u = np.broadcast_to(u, new_shape)
        else:
            u = np.random.rand(*new_shape)
        u = torch.Tensor(u)

    # Invert CDF
    u = u.contiguous()
    inds = torch.searchsorted(cdf, u, right=True)

    below = torch.max(torch.zeros_like(inds-1), inds-1)
    above = torch.min((cdf.shape[-1]-1) * torch.ones_like(inds), inds)
    inds_g = torch.stack([below, above], -1)  # (batch, N_samples, 2)

    matched_shape = [inds_g.shape[0], inds_g.shape[1], cdf.shape[-1]]
    cdf_g = torch.gather(cdf.unsqueeze(1).expand(matched_shape), 2, inds_g)
    bins_g = torch.gather(bins.unsqueeze(1).expand(matched_shape), 2, inds_g)
    T_g = torch.gather(T.unsqueeze(1).expand(matched_shape), 2, inds_g)
    tau_g = torch.gather(tau.unsqueeze(1).expand(matched_shape), 2, inds_g)

    ### Get tau diffs
    tau_diff = tau[...,1:] - tau[...,:-1]
    matched_shape_tau = [inds_g.shape[0], inds_g.shape[1], tau_diff.shape[-1]]

    tau_diff_g = torch.gather(tau_diff.unsqueeze(1).expand(matched_shape_tau), 2, below.unsqueeze(-1)).squeeze()

    s_left = bins_g[...,0]
    s_right = bins_g[...,1]
    T_left = T_g[...,0]
    tau_left = tau_g[...,0]
    tau_right = tau_g[...,1]

    dummy = torch.ones(s_left.shape, device=s_left.device)*-1.0
    samples1 = torch.where(torch.logical_and(tau_diff_g < zero_threshold, tau_diff_g > -zero_threshold), s_left, dummy)

    ### Our Precision Importance Sampling
    samples2 = torch.where(tau_diff_g >= zero_threshold, \
        pw_linear_sample(s_left, s_right, T_left, tau_left, tau_right, u, epsilon=epsilon_), samples1)

    samples = torch.where(torch.isnan(samples3), s_left, samples3)

    tau_g = torch.gather(tau.unsqueeze(1).expand(matched_shape), 2, inds_g)
    T_g = torch.gather(T.unsqueeze(1).expand(matched_shape), 2, inds_g)

    T_below = T_g[...,0]
    tau_below = tau_g[...,0]
    bin_below = bins_g[...,0]
    ####################################

    return samples, T_below, tau_below, bin_below

def pw_linear_sample(s_left, s_right, T_left, tau_left, tau_right, u, epsilon=1e-3):
    ln_term = -torch.log(torch.max(torch.ones_like(T_left)*epsilon, \
        torch.div(1-u, torch.max(torch.ones_like(T_left)*epsilon,T_left) ) ))
    discriminant = tau_left**2 + torch.div( 2 * (tau_right - tau_left) * ln_term , \
        torch.max(torch.ones_like(s_right)*epsilon, s_right - s_left) )

    t = torch.div( (s_right - s_left) * (-tau_left + torch.sqrt(\
        torch.max(torch.ones_like(discriminant)*epsilon, discriminant))) , \
    torch.max(torch.ones_like(tau_left)*epsilon, tau_right - tau_left))

    ### clamp t to [0, s_right - s_left]
    t = torch.clamp(t, torch.ones_like(t, device=t.device)*epsilon, s_right - s_left)

    sample = s_left + t

    return sample
```

Figure A5: **Code snippet for our Precise Importance Sampling.** The implementation of our precision importance sampling is also a direct replacement from the original function from the constant implementation called sample_pdf (See next figure for reference).

```python
### Original Piecewise Constant Opacity
def sample_pdf(bins, weights, N_samples, det=False, pytest=False):
    # Get pdf
    weights = weights + 1e-5 # prevent nans
    pdf = weights / torch.sum(weights, -1, keepdim=True)

    cdf = torch.cumsum(pdf, -1)
    cdf = torch.cat([torch.zeros_like(cdf[...,:1]), cdf], -1)  # (batch, len(bins))

    # Take uniform samples
    if det:
        u = torch.linspace(0., 1., steps=N_samples, device=bins.device)
        u = u.expand(list(cdf.shape[:-1]) + [N_samples])
    else:
        u = torch.rand(list(cdf.shape[:-1]) + [N_samples], device=bins.device)

    # Pytest, overwrite u with numpy's fixed random numbers
    if pytest:
        np.random.seed(0)
        new_shape = list(cdf.shape[:-1]) + [N_samples]
        if det:
            u = np.linspace(0., 1., N_samples)
            u = np.broadcast_to(u, new_shape)
        else:
            u = np.random.rand(*new_shape)
        u = torch.Tensor(u)

    # Invert CDF
    u = u.contiguous()

    inds = torch.searchsorted(cdf, u, right=True)

    below = torch.max(torch.zeros_like(inds-1), inds-1)
    above = torch.min((cdf.shape[-1]-1) * torch.ones_like(inds), inds)
    inds_g = torch.stack([below, above], -1)  # (batch, N_samples, 2)

    # cdf_g = tf.gather(cdf, inds_g, axis=-1, batch_dims=len(inds_g.shape)-2)
    # bins_g = tf.gather(bins, inds_g, axis=-1, batch_dims=len(inds_g.shape)-2)
    matched_shape = [inds_g.shape[0], inds_g.shape[1], cdf.shape[-1]]
    cdf_g = torch.gather(cdf.unsqueeze(1).expand(matched_shape), 2, inds_g)
    bins_g = torch.gather(bins.unsqueeze(1).expand(matched_shape), 2, inds_g)

    denom = (cdf_g[...,1]-cdf_g[...,0])
    denom = torch.where(denom<1e-5, torch.ones_like(denom), denom)
    t = (u-cdf_g[...,0])/denom
    samples = bins_g[...,0] + t * (bins_g[...,1]-bins_g[...,0])

    return samples
```

Figure A6: This is the original importance sampling for the constant approximation for reference.