# OpenReview forum: "NeRF Revisited: Fixing Quadrature Instability in Volume Rendering"
_NeurIPS.cc/2023/Conference — NeurIPS 2023 poster_

### Official Review · Reviewer_HvcE · 2023-06-27

**Soundness:** 3 good
**Presentation:** 2 fair
**Contribution:** 3 good
**Rating:** 3
**Confidence:** 4

**Summary:**

This paper describes a modification to the way NeRF performs approximate integration for volume rendering.  Instead of assuming piecewise constant opacity, they propose to use piecewise linear opacity.  They argue that this resolves what they dub the "quadrature instability" in NeRF.  They re-derive the quadrature integration method using piecewise linear opacity and piecewise constant color, and show that this also enables them to derive a precise inverse of the ray termination CDF for importance sampling.  Through experiments on standard NeRF datasets they show an quantitative improvement in rendering quality and also show qualitatively that their method makes the rendering sharper and more stable across change of viewpoint and camera distance.

**Strengths:**

This paper addresses an often overlooked aspect of NeRF which is the specifics of how the volume rendering is implemented and whether it could be improved.  The propose what to the best of my knowledge is a novel modification, which is to approximate the opacity along the ray as piecewise linear (instead of piecewise constant) in order to obtain a more accurate quadrature estimate of integral.  This formulation also leads them to derive a more precise method of importance sampling.

They analyze conceptually how the piecewise constant assumption in NeRF leads to conflicting ray supervision between perpendicular and grazing angle rays.  They claim that their piecewise linear formulation reduces this problem and leads to a more peaked opacity PDF.

They provide extensive derivations for the formulas in the supplemental material.  The new formulations are a "drop-in" replacement for the equations used in vanilla NeRF, and thus they can easily modify existing NeRF implementations to use them.

In their evaluation they compare against vanilla NeRF on the standard synthetic and real forward-facing datasets and show a quantitative and qualitative improvement in rendering quality across the board.

They also show that their method improves SCADE, a recent method that incorporates depth estimates into NeRF training, and show that swapping in their formulas leads to a modest improvement in quality.  They hypothesize that this is because of their more precise importance sampling.

**Weaknesses:**

They seemed to have missed a highly relevant previous work:

Wu, L., Lee, J. Y., Bhattad, A., Wang, Y. X., & Forsyth, D. (2022). DIVeR: Real-time and accurate neural radiance fields with deterministic integration for volume rendering. In Proceedings of the IEEE/CVF Conference on Computer Vision and Pattern Recognition (pp. 16200-16209).

This method is based on a voxel grid approach rather than an MLP.  However, the DIVeR paper also discusses the inaccuracy of integration using piecewise constant opacity.  Their solution allows for integration over more complicated functions than piecewise linear and so might be more accurate than the piecewise linear approach proposed here.  Indeed, DIVeR reports higher average rendering quality metrics on the NeRF synthetic dataset than what is reported here.

Because they claim that their method improves sampling and makes the opacity function more peaked at the surface, I would be very curious to see a comparison of surface reconstruction quality compared to NeuS:

Wang, P., Liu, L., Liu, Y., Theobalt, C., Komura, T., & Wang, W. (2021). NeuS: Learning Neural Implicit Surfaces by Volume Rendering for Multi-view Reconstruction. Advances in Neural Information Processing Systems, 34, 27171-27183.

The explanation of inverse transform sampling in the original NeRF method (section 3.2) should be expanded, as this is not explained thoroughly in the original NeRF paper.  Their discussion (L175-178) is quite brief considering how critical it is to the paper.

The paper has many grammatical errors and incomplete sentences which make it difficult to read.  I included some examples here:
* L135 "Hence the continuous probability density function (PDF) the ray r(s)"
* L138 "s is a point on the ray r" -> it doesn't make sense that s would be a point (since it is a scalar)
* L152-153 "P_j is the probability of each interval, which is mathematically equivalent to the probability of the interval." This sentence sounds like a meaningless tautology.
* L154 missing a period
* L117 "Then taking x = g^-1(u)." This is an incomplete sentence.
* L117-118 "However, this does not necessarily result in the samples from the actual ray distribution p(s) from the model."  This sentence doesn't work grammatically.  Some more analysis of why this doesn't result in sampling from the distribution would be helpful (explanations and visualizations).
* L203 "such an example of a sample-based loss used for NeRFs is depth"
* L232-3 "we pointed the drawback"
* L238 "thus the resulting CDF F being continuous"
* Table 1 caption: needs a period
* L273 capitalize Lego

**Questions:**

I think the most critical issue is the comparison with DIVeR.  How does this method compare to DIVeR, which appears to support an even more accurate computation of the integral?

I would also be interested to know whether they expect that this method will improve the surface reconstruction capability of NeRF, and how it compares to NeuS in this regard.

Finally I would like to know if they have a reference regarding the original NeRF's method of inverse transform sampling the CDF -- is this just based on the NeRF code or is there a reference for that?


**Limitations:**

A section on limitations does not appear in the paper.  A frank discussion of the limitations of the approach would strengthen the paper.

---

> ### Author Rebuttal · Authors · 2023-08-10
>
> Thank you for finding our formulation to be a novel modification. We address each of the concerns below.
>
> ## Q1: DIVeR citation
> Thank you for bringing up the paper. We will cite DIVeR in our revision.
>
> ## Q2 Comparison with DIVeR
> We plug our method into DIVeR by using their voxel-based representation and feature integration and dropping in our piecewise linear opacity formulation for volume rendering (PL-DIVeR). Results are shown in Table 2 demonstrating that our approach is on-par if not better across the different scenes in the Blender dataset. We use their official implementation and configuration for DIVeR64 at 128 voxels trained on a single Nvidia v100 GPU for each scene. We include this additional comparison in our revised version. We highlight that this shows the improvement of using our piecewise linear opacity formulation, which is a drop-in replacement to existing methods. Note that DIVeR is not directly comparable to PL-NeRF, but is instead compared against PL-DIVeR.
>
> ## Q3: Proof that DIVeR’s solution does not quite integrate over more complicated functions
> We note that technically it is not quite accurate to claim that “DIVeR’s solution allows for integration over more complicated functions”. We show that there is a tradeoff between the plausibility of the learned radiance field and their $MLP_w$ being an affine transformation and color being piecewise constant. Here we define a plausible radiance field as one where there is a unique opacity defined at each 3D point. We prove mathematically below that the volume rendering integral of DiVER does not integrate over more complicated functions if the underlying opacity field is plausible.
>
> ### Claim 1: DIVeR’s volume rendering equation holds if and only if color is piecewise constant.
> From Eq. 12 in DIVeR supp A.1., Hölder's inequality becomes equality iff color is constant along the ray in each voxel. The reverse direction is proven by substitution. The forward direction is shown by proving the contrapositive – if the radiance field is not constant in an open interval of the line segment, the integral of radiance over that open interval will be strictly smaller than the $L_\infty$ norm of radiance field over the same domain.
>
> ### Claim 2: A plausible density field, i.e. a unique opacity $\sigma(x) \forall x \in \mathbb{R}^3$, only exists if DIVeR’s $MLP_w$  is an affine transformation.
> We show this by showing the Hessian matrix of MLP_w is zero ranked. Let $S_1, … S_6$ be the six sides of a voxel and $S$ be the union of the sides. For $y \in S_i$, we define a function $$x \mapsto \int_0^{\lVert x-y\lVert}\sigma(r_{x, y}(t))dt=MLP_w(\int_{t^{in}}^{t^{out}}\hat{f}(r_{x, y}(t))dt) = MLP_w(\int_0^{\lVert x-y\lVert}\hat{f}(r_{x, y}(t))dt)$$,
> for $x \in S - S_i$, where equality follows from Eq. 5 in DIVeR main. By taking the gradient at $x \in S - S_i$ and rearranging the terms, we derive that $$\sigma(x)=x\cdot[\nabla MLP_w(\int_0^{\lVert x-y\lVert}\hat{f}(r_{x,y}(t))dt)]C(x).$$ Notice that $C(x)$ only depends on x. Since the RHS depends on $y \in S_i$ while the LHS does not, we can take the gradient of both sides w.r.t. to y and conclude the Hessian of $f \mapsto MLP_w(f)$ is rank zero on a convex open set. Then it follows that DIVeR’s $MLP_w$ is an affine transformation.
>
> Hence, there is a tradeoff between the plausibility of the learned radiance field and $MLP_w$ being an affine transformation, and under this scenario, it turns out to simply be a trilinear interpolation of opacity.
> ## Q4  Our opacity function being more peaked
> We quantitatively evaluate our precise importance sampling by taking our models trained with depth supervision. Specifically, we compute the average L2 distance between the ground truth depth and 64 random samples drawn from the fine networks using our precise importance sampling for PL-NeRF and the original inverse transform sampling for Vanilla NeRF. PL-NeRF and Vanilla NeRF attain an average error of 0.019 and 0.033, respectively, demonstrating that our approach with precise importance sampling draws samples closer to the ground truth surface.
>
> ## Q5 Improvement in geometry for NeRF
> We show quantitative improvement in our extracted geometry in Table 3. We extract the mesh from the learned opacity fields using marching cubes with a threshold of 25 and compute the distance between the ground truth model and the output mesh. Figure 2 also shows qualitative examples.
>
> We note that the SDF methods are orthogonal to our contribution as their goal is to learn an SDF field and convert that into density to enable supervision through volume rendering. How to render density is orthogonal to how density is outputted, our focus is the former, while the SDF works explore the latter. In principle, it is possible to use our method as a replacement for the volume rendering integration that these methods use. Exploring this could be an interesting direction for future work.
>
> ## Q6 Discussion on original NeRF’s inverse transform sampling
>
> Yes, it is based on the official codebase from the original NeRF that its successors are based on. Under the piecewise constant opacity assumption, the PDF of the ray termination distribution is not continuous, making its corresponding CDF also not continuous and thus non-invertible. Hence, to be able to sample with inverse transform sampling, a continuous surrogate CDF function (that we denoted as $G$) is needed, which from the official codebase is a linear interpolation. The code snippet is included in Figure S5 in our supplementary. Because a surrogate function is used, i.e. $G \neq \tilde{F}$, then the cumulative distribution described by $G$ is not equal to the cumulative distribution described by $\tilde{F}$, and hence the resulting samples from inverse transform sampling using $G$ is not equal to the samples drawn from the distribution, i.e. PDF $\tilde{f}$. We will elaborate on Sec 3.2 of the main paper in our revision.

---

> > ### Author Response · Authors · 2023-08-10
> >
> > Additionally, for limitations: A brief discussion of the limitations are found in our supplementary. Another additional limitation is we still assume piecewise color, i.e. within a bin we do not handle color integration. Modeling more sophisticated color integration can potentially handle other difficult scenarios such as double-walled colored glass or atmospheric effects such as fog or smoke. Thank you as well for pointing out the grammatical errors, and we will correct them in our revision.

---

> > ### Comment · Reviewer_HvcE · 2023-08-16
> >
> > Thank you for your responses.  I think the experimental results in the PDF are enough to show empirically that your method of volume integration is preferable to DIVER.  I am less sure about the proof you introduce here.  Fig. 5 of the DIVER paper shows how DIVER's MLP integration method can estimate the area under the curve of each segment more accurately than a constant or a piecewise-linear approximation.  Are you arguing in your proof that this does not actually happen when DIVER is applied in practice?  What if the "plausible density field" requirement is not met exactly but only approximately?

---

> > > ### Author Response · Authors · 2023-08-16
> > >
> > > Thank you very much for your response, especially for finding our experimental results convincing that our approach is preferable over DIVeR. The proof that we included in the rebuttal is to show that interestingly, DIVeR's integration method cannot actually integrate over more complicated functions, that is, their $MLP_w$ cannot approximate the integral of anything other than a trilinear interpolation.
> > >
> > > It seems that Figure 5 in the DIVeR paper is a schematic diagram accompanying their Sec. 4.2 that is used for illustrative purposes, rather than a demonstration of an actual experimental result. Yes, our proof does show that the right panel of Figure 5 cannot actually happen if there is a unique opacity defined at each 3D point. It turns out that (as shown in the proof) the only case when the output of their $MLP_w$ can be exact is when the integral is of a trilinear interpolation – this is a consequence from the proof that $MLP_w$ is an affine transformation. So if the function is any more complicated than that, their $MLP_w$ must either have large approximation errors, or their density field must become inconsistent, which would result in incorrect renderings from some other view.
> > >
> > > Yes, the "plausible density field" condition may not be exactly met, but the cost of that would be incorrect renderings from some other view, which is not desirable for novel view synthesis.
> > >
> > > We are happy to answer and clarify any further questions and concerns you may have.

---

> > > > ### Author Response · Authors · 2023-08-18
> > > >
> > > > Dear Reviewer HvcE,
> > > >
> > > > Please let us know if you have any further questions or concerns. We are happy to answer them and discuss further.
> > > >
> > > > Regards,
> > > > Authors

---

> > > > > ### Author Response · Authors · 2023-08-20
> > > > >
> > > > > Dear Reviewer HvcE,
> > > > >
> > > > > Just a friendly reminder that the discussion period is ending in less than 20 hours. If you still have any concerns, we would appreciate you responding as soon as possible, so that we can address them before the deadline. Otherwise, we would presume that all your concerns have been addressed, in which case we would appreciate you updating your rating. Thank you in advance.
> > > > >
> > > > > Regards,
> > > > >
> > > > > Authors

---

### Official Review · Reviewer_WoBC · 2023-07-04

**Soundness:** 3 good
**Presentation:** 4 excellent
**Contribution:** 3 good
**Rating:** 7
**Confidence:** 5

**Summary:**

Neural rendering methods like nerf relies on the integration of contributions (both color and density) along rays to predict views. Since analytical computation of the integral is not possible (the color and the density being both prediction from neural networks, usually MLPs), an approximation is estimated. Traditionally, the hypothesis done is that the color and the density are locally constant thus leading to a rectangle rule approximation. In this work, the authors propose to replace the the constant approximation of the density by a linear approximation. This allows for a more accurate estimation and a closed-form formula for sampling points during training.

**Strengths:**

The proposed modification is simple and elegant, it is mathematically driven leading to a closed-form formula. Moreover it is plug-and-play with most (all?) popular nerf-based methods and have a direct impact on the quality of the density estimation. It offers an important solution to the fuzzyness problem of the density learned by nerf methods, thus allowing for better surface reconstruction, and especially smaller details.

The paper is clear and well written. It introduces well all the different concepts necessary to understand the contribution.

Experimental results are convincing with a non-negligible  improvement of 0.5dB in PSNR but also across other metrics (SSIM and LPIPS) on multiple classic datasets (one synthetic and two real).

**Weaknesses:**

While the paper is very clear and the ideas elegant, I find the experimental section lacking:

* The proposed modification should impact mostly the density and thus the geometry of the reconstruction. The experimental section only focuses on a. While this is can be sufficient, I find disappointing that the authors did not take the opportunity to show the potential improvement of the geometry for the different datasets showed in the paper.

* The choice of baseline nerf is surprising. Given the type of experiments (with large changes of camera-to-scene distances), I think that mip-nerf would have made for a much better baselines. It already fixes a large number of artifacts pointed out by the authors in Figure S1 and therefore would have been a more appropriate comparison. I nonetheless expect that the conclusions can be transcribed to more recent frameworks.

* I find surprising that there is barely any discussion about the computation complexity of the proposed sampling (nor time measurements). I saw later that it is mentioned in a "Limitations" section in the supplementary but for me it's not a limitation per say and it should be studied in an ablation inside the paper. Especially since S1.1 shows that a better sampling can provide interesting convergence properties, thus allowing to change the number of samples used during the training. I would have expected a comparison with less samples than the 64 + 128 baseline (64 + 64 seems like an obvious thing to try based on the comments inside the text, I'm surprised that authors didn't even try that), to see the impact and find maybe a better compromise between computation time and performance. looking at the convergence plots would have also been interesting. Indeed, the proposed method might offer a faster convergence compared to the previous sampling, thus having a double impact.

* While slightly outside of the scope of this paper, it still would have been interesting to show the impact on the more practical methods (instant-ngp or voxel based methods for example) that are often preferred because of their fast training speed.

**Questions:**

RFF are more often referred to as LLFF. I suggest renaming to avoid any confusion (even though LLFF technically refers to the method proposed in the same paper) since a reader would expect that name.

**Limitations:**

Given the nature of the contributions, there's no major limitations nor potential negative societal impact to discuss. Some limitations are mentioned in the supplementary material but they are more related to the analysis of the method as I mentioned in "Weaknesses".

---

> ### Author Rebuttal · Authors · 2023-08-10
>
> Thank you for finding our work simple, very clear and well-written with elegant ideas that are mathematically driven. We appreciated that you highlighted that our approach is a plug-and-play to the popular nerf-based methods and that our experimental results are convincing with non-negligible improvement across metrics on multiple classes datasets. We address your questions below.
>
> ## Q1 Geometry Extraction
> Thank you for your suggestion. We extract the geometry from the learned density field from the trained models of PL-NeRF and Vanilla NeRF using marching cubes with a threshold of 25. Table 3 reports the distance between the surface of the ground truth model to the predicted meshes by sampling point clouds via ray casting. We see that our piecewise linear approach achieves a lower error compared to Vanilla NeRF on almost all the scenes in the Blender dataset. Figure 2-a shows qualitative results on the reconstruction of our piecewise linear vs the original piecewise constant formulation. As shown, we are able to better recover the holes on the body and wheels of the Lego scene as well as the interior structure inside the Mic. Moreover, interestingly, the numbers on the Drums scene is due to the surface of the drum being visually transparent as shown in the figure. We will include these results in our revision.
>
> ## Q2 Mip-NeRF baseline
> We plug-in our piecewise linear opacity for volume rendering into Mip-NeRF (PL-MipNeRF), and results are shown in Table 1. We demonstrate consistent improvement across all scenes on the Blender dataset using the standard train and test splits. We use the official released hyperparameters for MipNeRF and rerun their models using NeRFStudio. Our PL-MipNeRF uses the two-MLP training scheme with a coarse loss weight of 1.0, keeping all other hyperparameters fixed. Our results show that our modification is a drop-in replacement and can be transcribed to other more recent frameworks. Figure 1 shows qualitative examples where we see that under difficult scenarios such as when ray conflicts arise in the fine details of the Chair and in the presence of grazing angle views in the Mic, our PL-MipNeRF shows significant improvement over the baseline. We will include these results in our revised version.
>
> ## Q3 Computational complexity under different number of samples
> We measured the total rendering time for a single 800x800 image under different numbers of samples for our PL-NeRF. The total rendering time for (64+64), (64+128) and (128+64) are 19.20, 25.85 and 32.35 seconds, respectively. We will report these timings in our revision.
>
> ## Q4 Comparison with less samples
> Thank you for your suggestion. We run both our PL-NeRF and Vanilla NeRF with 64 coarse and 64 fine samples results in an average of (30.09, 0.939, 0.056) and (29.86, 0.937, 0.059) for (PSNR, SSIM, LPIPS), respectively, on the Blender dataset. This shows that under less number of samples our piecewise linear opacity formulation is better than the original piecewise constant opacity assumption. We will include this in our revision.
>
> ## Q5 Convergence plots under different number of samples
> Figure 2-b shows the convergence plots under different numbers of samples of our PL-NeRF vs Vanilla NeRF. We see that under different number of samples, our linear approach converges to a higher training PSNR.
>
> ## Q6 Additional results on a practical voxel-based method.
> We also demonstrate that our approach can also be integrated into a recent voxel-based method, DIVeR. We take its voxel-grid representation and feature integration and plug our piecewise linear opacity rendering formulation (PL-DIVeR). Table 2 shows quantitative results on the Blender dataset showing that even in voxel-based representations, our approach is on-par if not better than the piecewise constant opacity baseline. We will include these additional results in our revised version.
>
> ## Q7 Renaming RFF to LLFF
> Thank you for clarifying this and we will rename this in our revision to avoid confusion.

---

> > ### Author Response · Authors · 2023-08-19
> >
> > Thank you again for your time and effort in reviewing our paper. As the discussion phase is nearing to an end, please let us know if you have any further questions, and we will be more than happy to answer them.

---

> > ### Comment · Reviewer_WoBC · 2023-08-20
> >
> > Thanks for the additional information. The new results (with more baselines, different numbers of samples) reinforces my opinion. The proposal is not only interesting from a theoretical point of view but also has a clear practical impact (by only replacing the sampling part, something that can be done easily for most NeRF frameworks).

---

### Official Review · Reviewer_am3W · 2023-07-05

**Soundness:** 4 excellent
**Presentation:** 3 good
**Contribution:** 2 fair
**Rating:** 5
**Confidence:** 5

**Summary:**

This paper presents a fix for the "quadrature instability" arising from sampling of points for numerical integration used by NeRF2020 and its successors.The sampling inconsistence may not be big on classical rendering techniques, but can be significant when using neural architectures. Opacity and color values are assumed to be piecewise constant by NeRF. This paper offers a tractable solution with piecewise linear opacity values and piecewise constant color values. To avoid dense sampling the NeRF and follow up use a importance sampling approach based on corse-to-fine strategy. For drawing these importance samples the NeRF utilizes a surrogate CDF as original CDF is not invertible. The impact is shown on several standard scenes on difficult situations such as close to surfaces, fine feature areas, etc.


**Strengths:**

The proposed method is effective and efficient, drawing on existing volume-rendering literature. The modification suggested is rather simple but effective. The modified method should improve the quality of all Radiance Fields methods and hence of interest to researchers working on those areas.

**Weaknesses:**

The paper proposes a small, clean formulation of ray-integration that improves quality of NeRF reconstruction. The limited nature of the idea is its weakness, if it can be called so. The improvements in quantitative measures like PSNR is marginal, but the qualitative improvements are significant in particularly difficult situations. The usefulness and utility of the formulation could be established more clearly by incorporating it to other NeRF variants whose code is available. I would wholeheartedly support this idea in a vision conference like ICCV/CVPR. It is upto NeurIPS to decide if such a small-but-effective idea is of sufficient interest to its audience.

**Questions:**

Here are some questions/points:
- Can this method be integrated into all the other Radiance Field methods that essentially use the same ray-tracing formulation? Has it been done? That will increase its appeal
- What is the computational overhead of the expanded formulation compared to vanilla NeRF? The paper mentions 18 hours for 500K iterations. I am interested in knowing about head-to-head comparison with vanilla NeRF to know if there is significant performance penalty in using this formulation. I would like to know it for both training and rendering for view generation.
- Just curious: What is the adverse impact of using piecewise constant color values? What could the advantage be if we use linear color also? Does the other combination of piecewise constant density and piecewise linear color have similar closed-form solutions? What would the visual impact of that combination?
- Are comparisons with later NeRF such as ZipNeRF/MipNeRF available? It would be interesting to see if the new proposed strategy scales well to those as well.
- Depth supervised experiments seem valid. Does this pan-well against works like RGBDNeRF (Yuan et al. 2022 TPAMI)?



Minor points:
- Citations appear in ordinary parentheses as "(25)" instead of the more standard "[25]". This was very confusing in the beginning. Is this OK with NeurIPS style?
- Line 164: Isn't it more correctly R^3 x S^2 --> R^3? Why discretize it to [0-255]? Are you doing something different here?



**Limitations:**

What are the limitations of this approach in the authors' views? None are mentioned in the paper.

Showing the formulations effectiveness on other Radiance Field recovery methods (they all use the same rendering equation) will enhance the paper.

---

> ### Author Rebuttal · Authors · 2023-08-10
>
> Thank you for finding our formulation clean that results in a method that is simple but effective. We appreciate that you find our work to be of interest to researchers working in the area of radiance fields and for highlighting that our improvement is significant in particularly difficult situations. We answer the questions raised below.
>
> ## Q1 Comparison with Mip-NeRF
> We integrate our piecewise linear opacity formulation to the volume rendering integral into Mip-NeRF (PL-MipNeRF). Table 1 shows our quantitative results demonstrating consistent improvement across all scenes in the original hemisphere Blender dataset. Our experiments are run on the standard train and test split with the officially released hyperparameters of Mip-NeRF using the NerfStudio codebase. For PL-MipNeRF, we use the two-MLP training scheme with a coarse loss weight of 1.0. Figure 1 shows qualitative examples where we see that under difficult scenarios such as when ray conflicts arise in the fine details of the Chair and in the presence of grazing angle views in the Mic, our PL-MipNeRF shows significant improvement over the baseline. Our results show that our piecewise linear opacity and piecewise constant color formulation scales well to Mip-NeRF as well. We will include this in our revision.
>
> ## Q2 Formulation’s effectiveness to other radiance fields methods
> We also demonstrate our formulation’s effectiveness on other radiance field methods, such as Mip-NeRF as presented above, and additionally DIVeR, a recent voxel–grid method. We plug our method into their approach by utilizing their voxel-based representation and feature integration and dropping in our piecewise linear opacity rendering formulation (PL-DIVeR). We similarly run the DIVeR models on a single Nvidia v100 GPU trained using their default configurations and hyperparameters for DIVeR64 at 128 voxels. Table 2 shows the quantitative results on the standard Blender dataset, showing that our formulation is an effective drop-in replacement in other radiance field methods. We will include these results in our revision.
>
> ## Q3 Computational overhead compared to Vanilla NeRF
> The total training time for 500k iterations on a single Nvidia v100 GPU is 17.78 and 21.43 hours for Vanilla NeRF and PL-NeRF, respectively. Figure 2-c shows the head-to-head comparison of training PSNR (y-axis) with respect to time (x-axis) of Vanilla NeRF vs PL-NeRF on the Lego scene. Rendering a single 800x800 image takes 25.59 and 32.35 seconds for Vanilla NeRF and PL-NeRF, respectively. We will include these findings and specify the training and rendering time in our revision.
>
> ## Q4 Possible advantage of piecewise linear color; impact of using piecewise constant color
> An example scenario where piecewise linear color could have an advantage is a scene with a thin glass (or some non-opaque surface) with one color, e.g. red, on one side and another color, e.g. blue, on the other. Piecewise linear color would smoothly blend the color of the two sides.
>
> ## Q5 Would piecewise linear color and piecewise constant opacity have closed-form solutions? Impact of such combinations.
> Yes, it is closed-form. Under piecewise constant opacity and piecewise linear color, we can write the expected color of the bin as $\int_{s_i}^{s_{i+1}} \tau(s)T(s)c(s)ds = T(s_i)\tau_i \int_{s_i}^{s_{i+1}} c(s) \exp{(-\tau_i(s-s_i))}ds$ (refer to Sec 2.3 from Max and Chen 2010). This expression is of the form $\int A(s)\exp{B(s)}ds$, where $A(s)$ and $B(s)$ are both linear functions, hence its integral is in closed form using integration by parts. This assumption can potentially tackle the situation presented above on the double-colored glass, however, it still has a problem for the scenarios we have presented in the paper such as grazing angle and cameras at different distances.
>
> ## Q6 Marginal improvements in quantitative measures
> Our results show significant qualitative improvements in particularly difficult situations as mentioned in your review. Unfortunately, the standard metrics are not designed to be sensitive to those specific scenarios, but nonetheless, we point out that Reviewer WoBC states that our experimental results are convincing with a “non-negligible improvement” across metrics in multiple classic datasets. We also note that as with any other rendering methods, if the underlying simpler assumption suffices, e.g. piecewise constant opacity, suffices for a particular scene, then the improvement of the more sophisticated model, e.g. piecewise linear opacity, is less apparent.
>
> ## Q7 RGBDNeRF
> RGBDNeRF is orthogonal to our method as it depth-supervised experiments as it does not use a depth-based loss. As mentioned by HvcE, the results of our depth-supervised experiments demonstrate that our piecewise linear approach improves on the sample-based depth loss of a recent method, SCADE, compared to the original piecewise constant approximation.
>
> ## Q8 Citation format
> Thank you. We will change the citation format to the more standard “[25]” in our revision.
>
> ## Q9 Ln 164
> Thanks for catching this technicality. We are not doing anything different here – we will clarify and correct this in our revision.
>
> ## Q10 Limitations
> One potential limitation is our piecewise color assumption, i.e. within a bin we do not integrate color. This assumption will struggle in difficult scenarios such as two-sided colored glass or atmospheric effects such as fog or smoke. We will add this limitation in our revision. However, we also note a statement from Reviewer WoBC that “given the nature of the contributions, there are no major limitations nor potential negative impacts to discuss.”

---

> > ### Author Response · Authors · 2023-08-19
> >
> > Thank you again for your time and effort in reviewing our paper. As the discussion phase is nearing to an end, please let us know if you have any further questions and we will be more than happy to answer them.

---

### Official Review · Reviewer_tVLL · 2023-07-07

**Soundness:** 3 good
**Presentation:** 3 good
**Contribution:** 3 good
**Rating:** 6
**Confidence:** 4

**Summary:**

The paper addresses a fundamental limitation of existing NeRF formulation, namely piecewise constant integration, which results in sensitivity to point sampling, summarized as quadrature instability in this paper. To address this, the paper proposes a new formulation based on piecewise linear quadrature for density and piecewise constant quadrature for color. The paper demonstrates that the proposed method produces sharper and more stable results, and leads to better quantitative metrics.

**Strengths:**

The paper clearly analyzes the problem of existing methods.

The proposed solution is theoretically sound.

The proposed method leads to improvements in NeRF reconstruction, both qualitatively and quantitatively.

The paper shows detailed theoretical analysis and visual comparisons in supplementary material and video.

**Weaknesses:**

The improvements appear to be quite small from the overall quantitative metrics.

It seems that the improvements may be more visible in specific scenarios, but may be subtle for general inputs.

**Questions:**

Is visual difference obvious for a random input, or if the example shown needs to be carefully chosen?


**Limitations:**

The paper does not discuss its limitations. It may happen that the method is robust, but some limitations (including limited improvements) in certain scenarios should still be discussed.

---

> ### Author Rebuttal · Authors · 2023-08-10
>
> Thank you for finding our paper clear and theoretically sound that addresses a fundamental problem of the existing NeRF formulation leading to quantitatively and qualitatively improvements in NeRF reconstruction. Below we address the questions that were raised.
>
> ## Q1 Small improvements from overall quantitative metrics.
> As emphasized by Reviewer WoBC, our “experiment results are convincing with a non-negligible improvement of 0.5dB in PSNR but also across other metrics (SSIM, LPIPS) on multiple classic datasets (one synthetic and two real).” One advantage of our approach is highlighted in difficult scenarios where sensitivity to samples becomes a problem from the original piecewise constant formulation. The metrics are not designed to be sensitive to those specific scenarios, but as also mentioned by Reviewer am3W the “qualitative improvements are significant in particularly difficult scenarios”.
>
> ## Q2 Visual difference of random views
> In scenes with difficult scenarios as mentioned in the paper, random views will have visual differences. On the other hand, in cases where the piecewise constant opacity assumption suffices, the performance is similar – as with any other rendering methods, if the underlying assumption of a simpler rendering model suffices, then the improvement of the more sophisticated model is less apparent.
>
> ## Q3 Limitations
> One potential limitation is we still assume piecewise color. Enabling a more sophisticated color function can potentially handle difficult scenarios such as double-walled colored glass or atmospheric effects such as fog or smoke. We will add this to our revision, however, we also note a statement from Reviewer WoBC that “given the nature of the contributions, there are no major limitations nor potential negative impacts to discuss.”

---

> > ### Comment · Reviewer_tVLL · 2023-08-18
> >
> > Thanks for your clarification.

---

> > > ### Author Response · Authors · 2023-08-19
> > >
> > > You’re welcome. Let us know if you have any further questions.

---

### Author Rebuttal · Authors · 2023-08-10

We thank the reviewers for their feedback and for finding our approach novel (HvcE), simple (WoBC, am3W) and effective (am3W) that introduces a clean (am3W), elegant (WoBC) and theoretically sound (tVLL) formulation that is clear and well-written (WoBC). As summarized by the reviewers, we tackle a commonly overlooked problem for NeRFs on how integration is done for volume rendering. Specifically, we propose a piecewise linear opacity and piecewise constant color formulation to volume rendering that alleviates quadrature instability and leads to a simple and closed-form equation. Our new formulation is a plug-and-play (WoBC) or drop-in replacement (HvcE) to existing NeRF-based methods that assumes piecewise constant opacity and color. Our experiments show that this leads to quantitative improvements across different metrics on multiple classic datasets
(WoBC) that is qualitatively significant in particularly difficult situations (am3W). Below we include a brief summary to each response. Please see individual reviewer responses for more details.

## [tVLL, am3W] Marginal quantitative improvement
As mentioned by WoBC, our experiment results “are convincing with a non-negligible improvement across metrics on multiple datasets,” and the improvements of our approach are highlighted in “particularly difficult situations” (am3W). We note that the metrics are not designed to be sensitive to those specific scenarios.

## [tVLL] Visual difference of random inputs
Random inputs will have visual differences in scenes with difficult scenarios as mentioned in the paper. However, as with any other rendering methods, if the underlying simpler assumption, e.g. piecewise constant, suffices, then the improvement of the more sophisticated model, e.g. piecewise linear, is less apparent.

## [am3W, WoBC] Comparison with Mip-NeRF
We plug-in our piecewise linear opacity formulation into Mip-NeRF (PL-MipNeRF). Table 1 shows consistent improvement across all scenes in all metrics on the Blender dataset, demonstrating that results can scale to more recent frameworks.

## [am3W, WoBC] Integration to other methods
In addition to MipNeRF, we also integrate our approach into DIVeR, a recent voxel-based approach (PL-DIVeR), as shown in Table 2. Results show that our formulation can be utilized as a drop-in replacement to the recent voxel-based approach.

## [am3W] Computation overhead compared to Vanilla NeRF
On a single Nvidia V100 GPU, the total training time for 500k iterations of Vanilla-NeRF and PL-NeRF is 17.78 and 21.43 hours, respectively, while the total rendering time for one 800x800 image is 25.59 and 32.35 seconds, respectively. Figure 2-c shows the training curves with respect to time for the Lego scene.

## [am3W] Possible advantage of linear color
An example scene where this can be advantageous is a thin glass with one color on one side and a different color on the other.

## [am3W] Will there be a similar closed-form expression for piecewise linear color and piecewise constant opacity?
Yes, it turns out that it is in closed-form too. However, that assumption will still have the problems for the difficult scenarios presented in our paper.

## [am3W] RGBDNeRF
RGBDNeRF is orthogonal to our experiments on depth loss with our piecewise linear formulation as this work does not supervise their NeRF training with depth.

## [WoBC] Geometry Extraction
Table 3 shows the average distance between the surface of the ground truth model to the extracted meshes of the trained models for Vanilla NeRF and PL-NeRF. Figure 2-a shows qualitative examples.

## [WoBC] Computational complexity of different number of samples
The total rendering time for a 800x800 image of PL-NeRF is 19.20, 25.85, 32.35 seconds for (64+64), (64+128) and (128+64) samples, respectively.

## [WoBC] Less number of samples (64+64)
The results for PL-NeRF and Vanilla NeRF for (64+64) samples is (30.09, 0.939, 0.056) and (29.86, 0.937, 0.059), respectively, for (PSNR, SSIM, LPIPS).

## [WoBC] Convergence plots
Figure 2-b shows the convergence plots under different number of samples for PL-NeRF and Vanilla NeRF.

## [HvcE] DIVeR citation
We will cite DIVeR in our revision.

## [HvcE] Comparison with DIVeR
We incorporate our approach to DIVeR by using their voxel representation and feature integration and plugging in our piecewise linear opacity volume rendering integration. Table 2 shows quantitative results.

## [HvcE] DIVeR does not quite allow for integration over more complicated functions.
Interestingly, we show that there is a tradeoff between the plausibility of the learned radiance field and DIVeR’s $MLP_w$ being an affine transformation and color being piecewise constant. As a result, it does not quite allow for the integration over more complicated functions.

## [HvcE] Method improves sampling making it more peaky at the surface
We quantitatively evaluate the samples drawn from the learned distributions of PL-NeRF and Vanilla NeRF by computing the average L2 distance between the samples and the ground truth depth. The results are 0.019 and 0.033 for PL-NeRF and Vanilla NeRF, respectively.

## [HvcE] Improvement in geometry for NeRF
Table 3 shows the improvement in the extracted geometry of PL-NeRF compared to Vanilla NeRF across the different scenes in the Blender dataset. Figure 2-a shows qualitative examples. We note that the SDF methods are orthogonal to our contribution as their goal is to learn an SDF field and convert that into density to enable supervision through volume rendering. How to render density is orthogonal to how density is outputted, our focus is the former, while the SDF works explore the latter.

## [HvcE] Reference for inverse transform sampling of the original NeRF
Yes, it is based on their official codebase where linear interpolation is used as the surrogate function $G$. Since a surrogate function is used then the samples are not being drawn from the original underlying distribution.

---

### Decision · Program_Chairs · 2023-09-21

**Decision:**

Accept (poster)

**Comment:**

The paper presents a method for improving the stability of the volume rendering process commonly used in the literature. Most reviewers are positive about this paper, given the simplicity of the technique and its significance to the community. The rebuttal results showcase the wider applicability of the proposed sampling methods, which reviewers appreciate. However, there were concerns about the complicated proof regarding the "plausible density field". Given the promising empirical results, the reviewers agree that this paper has sufficient merits to be accepted but are uncomfortable with having the proof in the camera ready since these theoretical justifications must be properly reviewed before it can be included in the paper.

Considering the suggestions from the reviewers, the AC decides to accept the paper, but urges the authors *not* to include additional proof in the final submission.